# If MaxEnt RL is the Answer,
# What is the Question?

## Abstract

Experimentally, it has been observed that humans and animals often make decisions that do not maximize their expected utility, but rather choose outcomes randomly, with probability proportional to expected utility. Probability matching, as this strategy is called, is equivalent to maximum entropy reinforcement learning (MaxEnt RL). However, MaxEnt RL does not optimize expected utility. In this paper, we formally show that MaxEnt RL does optimally solve certain classes of control problems with variability in the reward function. In particular, we show (1) that MaxEnt RL can be used to solve a certain class of POMDPs, and (2) that MaxEnt RL is equivalent to a two-player game where an adversary chooses the reward function. These results suggest a deeper connection between MaxEnt RL, robust control, and POMDPs, and provide insight for the types of problems for which we might expect MaxEnt RL to produce effective solutions. Specifically, our results suggest that domains with uncertainty in the task goal may be especially well-suited for MaxEnt RL methods.

## 1 Introduction

Reinforcement learning (RL) searches for a policy that maximizes the expected, cumulative reward. In fully observed Markov decision processes (MDPs), this maximization always has a deterministic policy as a solution. Maximum entropy reinforcement learning (MaxEnt RL) is a modification of the RL objective that further adds an entropy term to the objective. This additional entropy term causes MaxEnt RL to seek policies that (1) are stochastic, and (2) have non-zero probability of sampling every action. MaxEnt RL can equivalently be viewed as *probability matching* between trajectories visited by the policy and a distribution defined by exponentiating the reward (See Section 2). MaxEnt RL has appealing connections to probabilistic inference (Dayan & Hinton, 1997; Neumann et al., 2011; Todorov, 2007; Kappen, 2005; Toussaint, 2009; Rawlik et al., 2013; Theodorou et al., 2010; Ziebart, 2010), prompting a renewed interest in recent years (Haarnoja et al., 2018b; Abdolmaleki et al., 2018; Levine, 2018). MaxEnt RL can also be viewed as using Thompson sampling (Thompson, 1933) to collect trajectories, where the posterior belief is given by the exponentiated return. Empirically, MaxEnt RL algorithms achieve good performance on a number of simulated (Haarnoja et al., 2018b) and real-world (Haarnoja et al., 2018a; Singh et al., 2019) control tasks, and can be more robust to perturbations (Haarnoja et al., 2018c).

There is empirical evidence that behavior similar MaxEnt RL is used by animals in the natural world. While standard reinforcement learning is often used as a model for decision making (Scott, 2004; Liu & Todorov, 2007; Todorov & Jordan, 2002), many animals, including humans, do not consistently make decisions that maximize expected utility. Rather, they engage in *probability matching*, choosing actions with probability proportional to how much utility that action will provide. Examples include ants (Lamb & Ollason, 1993), bees (Greggers & Menzel, 1993), fish (Bitterman et al., 1958), ducks (Harper, 1982), pigeons (Bullock & Bitterman, 1962; Graf et al., 1964), and humans, where it has been documented so extensively that Vulkan (2000) wrote a survey of surveys of the field. This effect has been observed not just in individuals, but also in the collective behavior of groups of animals (see Stephens & Krebs (1986)), where it is often described as obtaining the ideal free distribution. Probability matching is not merely a reflection of youth or ignorance. Empirically, more intelligent creatures are more likely to engage in probability matching. For example, in a comparison of Yale students and rats, Gallistel (1990) found that the students nearly always performed probability

matching, while rats almost always chose the maximizing strategy. Similarly, older children and adults engage in probability matching more frequently than young children (Stevenson & Odom, 1964; Weir, 1964). While prior work has offered a number of explanations of probability matching (Vulkan, 2000; Gaissmaier & Schooler, 2008; Wozny et al., 2010; Sakai & Fukai, 2008), its root cause remains an open problem.

The empirical success of MaxEnt RL algorithms on RL problems is surprising, as MaxEnt RL optimizes a different objective than standard RL. The solution to every MaxEnt RL problem is stochastic, while deterministic policies can always be used to solve standard RL problems (Puterman, 2014). While RL can be motivated from the axioms of utility theory (Russell & Norvig, 2016), MaxEnt RL has no such fundamental motivation. It remains an open question as to whether the standard MaxEnt RL objective actually optimizes some well-defined notion of risk or regret that would account for its observed empirical benefits. This paper studies this problem, and aims to answer the following question: *if MaxEnt RL is the solution, then what is the problem?* Answering this question is a first step towards understanding the empirical success of MaxEnt RL algorithms, and our analysis will suggest that MaxEnt RL might be applicable to problems typically considered to be much more complex than standard RL.

In this paper, we show that MaxEnt RL provides the optimal control solution in settings with uncertainty and variability in the reward function. More precisely, we show that MaxEnt RL is equivalent to two more challenging problems: (1) regret minimization in a *meta-POMDP*, and (2) *robust-reward control*. The first setting, the meta-POMDP, is a partially observed MDP where the reward depends on an unobserved portion of the state, and where multiple episodes in the original MDP correspond to a single extended trial in the meta-POMDP. While seemingly Byzantine, this type of problem setting arises in a number of real-world settings discussed in Section 3. Optimal policies for the meta-POMDP must explore at test-time, behavior that cannot result from maximizing expected utility. In the second setting, robust-reward control, we consider an adversary that chooses some aspects of the reward function. Intuitively, we expect stochastic policies to be most robust because they are harder to exploit, as we formalize in Section 5. Even if the agent will eventually be deployed in a setting without adversaries, the adversarial objective bounds the worst-case performance of that agent. Our result in this setting can be viewed as an extension of prior work connecting the principle of maximum entropy to two-player games (Ziebart et al., 2011; Grünwald et al., 2004). While both robust-reward control and regret minimization in a meta-POMDP are natural problems that arise in many real-world scenarios, neither is an expected utility maximization problem, so we cannot expected optimal control to solve these problems. In contrast, we show that MaxEnt RL provides solutions to both. In summary, our analysis suggests that the empirical benefits of MaxEnt RL arise by implicitly solving control problems with variability in the reward.

## 2 PRELIMINARIES

We begin by defining notation and discussing some previous motivations for MaxEnt RL. An agent observes states $s_t$, takes actions $a_t \sim \pi(a_t \mid s_t)$, and obtains rewards $r(s_t, a_t)$. The initial state is sampled $s_1 \sim p_1(s_1)$, and subsequent states are sampled $s' \sim p(s' \mid s, a)$. Episodes have $T$ steps, which we summarize as a trajectory $\tau \triangleq (s_1, a_1, \cdots, s_T, a_T)$. Without loss of generality, we can assume that rewards are undiscounted, as any discount can be addressed by modifying the dynamics to transition to an absorbing state with probability $1 - \gamma$. The RL objective is:

$$\arg\max_{\pi} \mathbb{E}_{\pi}\left[\sum_{t=1}^{T} r(s_t, a_t)\right] = \int \left(\sum_{t=1}^{T} r(s_t, a_t)\right) p_1(s_1) \prod_{t=1}^{T} p(s_{t+1} \mid s_t, a_t)\pi(a_t \mid s_t)d\tau.$$

In fully observed MDPs, there always exists a deterministic policy as a solution (Puterman, 2014). The MaxEnt RL problem, also known as the entropy-regularized control problem, is to maximize the sum of expected reward and conditional action entropy, $\mathcal{H}_{\pi}[a \mid s]$:

$$\arg\max_{\pi} \mathbb{E}_{\pi}\left[\sum_{t=1}^{T} r(s_t, a_t)\right] + \mathcal{H}_{\pi}[a \mid s] = \mathbb{E}_{\pi}\left[\sum_{t=1}^{T} r(s_t, a_t) - \log \pi(a_t \mid s_t)\right]$$

The MaxEnt RL objective results in policies that are stochastic, with higher-entropy action distributions in states where many different actions lead to similarly optimal rewards, and lower-entropy

distributions in states where a single action is much better than the rest. Moreover, MaxEnt RL results in policies that have non-zero probability of sampling any action. MaxEnt RL can equivalently be defined as a form of probability matching, minimizing a reverse Kullback Leibler (KL) divergence (Rawlik et al., 2013):

$$\max_\pi \mathbb{E}_\pi \left[ \sum_{t=1}^T r(s_t, a_t) \right] + \mathcal{H}_\pi[a \mid s] = -\min_\pi D_{\text{KL}}(\pi(\tau) \parallel p_r(\tau)),$$

where the policy distribution $\pi(\tau)$ and the target distribution $p_r(\tau)$ are defined as

$$p_r(\tau) \propto p_1(s_1) \prod_{t=1}^T p(s_{t+1} \mid s_t, a_t) e^{\sum_{t=1}^T r(s_t, a_t)}, \quad \pi(\tau) \triangleq p_1(s_1) \prod_{t=1}^T p(s_{t+1} \mid s_t, a_t) \pi(a_t \mid s_t).$$

Prior work on MaxEnt RL offers a slew of intuitive explanations for why one might prefer MaxEnt RL. We will summarize three common explanations and highlights problems with each.

**Exploration**: MaxEnt RL is often motivated as performing good exploration. Unlike many other RL algorithms, such as DQN (Mnih et al., 2015) and DDPG (Lillicrap et al., 2015), MaxEnt RL performs exploration and policy improvement with the same (stochastic) policy. One problem with this motivation is that stochastic policies can be obtained directly from standard RL, without adding an entropy term (Heess et al., 2015). More troubling, while MaxEnt RL learns a stochastic policy, many MaxEnt RL papers evaluate the corresponding deterministic policy (Haarnoja et al., 2018b), suggesting that the stochastic policy is not what should be optimized. While improved exploration may be an ancillary benefit of MaxEnt RL, it remains unclear why we should expect MaxEnt RL to explore better than standard RL algorithms that learn stochastic policies.

**Probabilistic inference**: Connections with probabilistic inference offer a second motivation for MaxEnt RL (Abdolmaleki et al., 2018; Haarnoja et al., 2018b; Todorov, 2007; Levine, 2018; Toussaint, 2009). These approaches cast optimal control as an inference problem by defining additional optimality binary random variables $\mathcal{O}_t$, equal one with probability proportional to exponentiated reward. These methods then maximize the following log-likelihood:

$$\log p(\mathcal{O}_t) = \log \int p_1(s_1) \prod_{t=1}^T p(\mathcal{O}_t = 1 \mid s_t, a_t) p(s_{t+1} \mid s_t, a_t) \pi(a_t \mid s_t) d\tau$$

$$= \log \int p_1(s_1) e^{\sum_{t=1}^T r(s_t, a_t)} \prod_{t=1}^T p(s_{t+1} \mid s_t, a_t) \pi(a_t \mid s_t) d\tau$$

$$= \log \mathbb{E}_\pi \left[ e^{\sum_{t=1}^T r(s_t, a_t)} \right] \approx \mathbb{E}_\pi \left[ \sum_{t=1}^T r(s_t, a_t) \right] + \text{Var}_\pi \left[ \sum_{t=1}^T r(s_t, a_t) \right] \quad (1)$$

The last term is a cumulant generating function (i.e., the logarithm of a moment generating function (Gut, 2013, Chpt. 6)), which can be approximated as the sum of expected reward and variance of returns (Mihatsch & Neuneier, 2002). Thus, directly maximizing likelihood leads to risk-seeking behavior, not optimal control. Equation 1 can also be directly obtained by considering an agent with a risk-seeking utility function (O'Donoghue, 2018). While risk seeking behavior can be avoided by maximizing a certain lower bound on Equation 1 (Levine, 2018), artificially constraining algorithms to maximize a lower bound suggests that likelihood is not what we actually want to maximize.

**Easier optimization:** Finally, some prior work (Ahmed et al., 2018; Williams & Peng, 1991) argues that the entropy bonus added by MaxEnt RL makes the optimization landscape smoother. However, it does not suggest why optimizing the wrong but smooth problem yields a good solution to the original optimization problem.

## 3 What Problems Does MaxEnt RL Solve?

MaxEnt RL produces stochastic policies, so we first discuss when stochastic policies may be optimal. Informally, the two strengths of stochastic policies are that they (1) are guaranteed to eventually try every action sequence and (2) do not always choose the same sequence of actions.

### 3.1 ANSWER 1: PARTIALLY OBSERVED ENVIRONMENTS

The first strength of stochastic policies guarantees that they will not have to wait infinitely long to find a good outcome. Imagine that a cookie is hidden in one of two jars. A policy that always chooses to look in the same jar (say, the left jar) may never find the cookie if it is hidden in the other jar (the right jar). Such a policy would incur infinite regret. This need to try various approaches arises in many realistic settings where we do not get to observe the true reward function, but rather have a belief over what the true reward is. For example, in a health-care setting, consider the course of treatment for a patient. The desired outcome is to cure the patient. However, whether the patient is cured by different courses of treatment depends on their illness, which is unknown. A physician will prescribe medications based on his beliefs about the patient's illness. If the medication fails, the patient returns to the physician the next week, and the physician recommends another medication. This process will continue until the patient is cured. The physician's aim is to minimize the number of times the patient returns. Another example is a robot that must perform chores in the home based on a user's commands. The true goal in this task is to satisfy the user. Their desires are never known with certainty, but must be inferred from the user's behavior. Indeed, arguably the majority of problems to which we might want to apply reinforcement learning algorithms are actually problems where the true reward is unobserved, and the reward function that is provided to the agent represents an imperfect belief about the goal. In Section 4, we define a meta-level POMDP for describing these sorts of tasks and show that MaxEnt RL minimizes regret in such settings.

### 3.2 ANSWER 2: ROBUSTNESS TO EXPLOITATION

The second strength of stochastic policies is that they are harder to exploit. For example, in the game rock-paper-scissors ("ro-sham-bo"), it is bad to always choose the same action (say, rock) because an adversary can always choose an action that makes the player perform poorly (e.g., by choosing paper). Indeed, the Nash existence theorem (Nash et al., 1950) requires stochastic policies to guarantee that a Nash equilibrium exists. In RL, we might likewise expect that a randomized policies are harder to exploit than deterministic policies. To formalize the intuition that MaxEnt policies are robust against adversaries, we define the robust-reward control problem.

**Definition 3.1.** The *robust-reward control problem* for a set of reward functions $R = \{r_i\}$ is

$$\arg \max_{\pi} \min_{r' \in R} \mathbb{E}_{\pi} \left[ \sum_{t=1}^{T} r'(s_t, a_t) \right].$$

We can think of this optimization problem as a two-player, zero-sum game between a *policy player* and an adversarial *reward player*. The policy player chooses the sequence of actions in response to observations, while the reward player chooses the reward function against which the states and actions will be evaluated. This problem is slightly different from typical robust control (Zhou & Doyle, 1998), as it considers perturbations to rewards, not dynamics. Typically, solving the robust-reward control problem is challenging because it is a saddle-point problem. Nonetheless, in Section 5, we show that MaxEnt RL is exactly equivalent to solving a robust-reward control problem.

### 3.3 SUMMARY

Together, these two properties suggest that stochastic policies, such as those learned with MaxEnt RL, can be robust to variability in the reward function. This variability may be caused by (1) a designer's uncertainty about what the right reward should be, (2) the presence of perturbations to the reward (e.g., for an agent that interacts with human users, who might have different needs and wants in each interaction), or (3) partial observability (e.g., a robot in a medical setting may not observe the true cause for a patient's illness). In this paper, we formally show that MaxEnt RL algorithms produce policies that are robust to two distinct sources of reward variability: unobserved rewards in partially observed Markov decision processes (POMDPs) and adversarial variation in the rewards.

## 4 MAXIMUM ENTROPY RL AND PARTIALLY OBSERVED ENVIRONMENTS

In this section, we formalize the intuition from Section 3 that stochastic policies are preferable in settings with unknown tasks. We first describe the problem of solving an unknown task as a special

class of POMDPs, and then show that MaxEnt RL provides the optimal solution for these POMDPs. Our results in this section suggest a tight coupling between MaxEnt RL and regret minimization.

We begin by defining the *meta-POMDP* as a MDP with many possible tasks that could be solved. Solving a task might mean reaching a particular goal state or performing a particular sequence of actions. We will use the most general definition of success as simply matching some target trajectory, $\tau^*$. Crucially, the agent does not know which task it must solve (i.e., $\tau^*$ is not observed). Rather, the agent has access to a belief $p(\tau)$ over what the target trajectory may be. This results in a POMDP, where the agent's ignorance of the true task makes the problem partial observed.

Each *meta-step* of the meta-POMDP corresponds to one episode of the original MDP. A *meta-episode* is a sequence of meta-steps, which ends when the agent solves the task in the original MDP. Intuitively, each meta-episode in the meta-POMDP corresponds to multiple trials in the original MDP, where the task remains the same across trials. The agent keeps interacting with the MDP until it solves the task. While the meta-POMDP might seem counter-intuitive, it captures many practical scenarios. For example, in the health-care setting in Section 3, the physician does not know the patient's illness, and may not even know when the patient has been cured. Each meta-step corresponds to one visit to the physician, which might entail running some tests, performing an operation, and prescribing a new medication. The meta-episode is the sequence of patient visits, which ends when the patient is cured. As another example, Appendix A.2 describes how meta-learning can also be viewed as a meta-POMDP.

Before proceeding, we emphasize that defining the meta-POMDP in terms of trajectory distributions is strictly more general than defining it in terms of state distributions. However, Section 4.3 will discuss how goal-reaching, a common problem setting in current RL research (Lee et al., 2019b; Warde-Farley et al., 2018; Pong et al., 2019), can be viewed as a special case of this general formulation.

## 4.1 REGRET IN THE META-POMDP

The meta-POMDP has a simple reward function: $+1$ when the task is completed, and $0$ otherwise. Since the optimal policy would solve the task immediately, its reward on every meta-step would be one. Therefore, the regret is given by $1 - 0 = 1$ for every meta-step when the agent fails to solve the task, and $1 - 1 = 0$ for the (final) meta-step when the agent solves the task. Thus, the cumulative regret of an agent is the expected number of meta-steps required to complete the task. For example, in the health-care example, the regret is the number of times the patient visits the physician before being cured.

Our analysis of the meta-POMDP will consider policies that are Markovian within a meta-episode: while the policy can be updated between meta-episodes, the policy cannot use information from one meta-step to take better actions in a future meta-step within the same meta-episode. Our results will therefore be lower bounds on the performance of non-Markovian policies. The Markovian assumption is equivalent to saying that agents lack memory, and might be seen as an instantiation of bounded rationality. That MaxEnt RL is optimal under this assumption suggests that the probability matching observed in nature is optimal under memory constraints.

Mathematically, we use $\pi(\tau^*)$ to denote the probability that policy $\pi$ produces target trajectory $\tau^*$. Then, the number of episodes until it matches trajectory $\tau^*$ is a geometric random variable with parameter $\pi(\tau^*)$. The expected value of this random variable is $1/\pi(\tau^*)$, so we can write the regret of the meta-POMDP as:

$$\text{Regret}_p(\pi) = \mathbb{E}_{\tau^* \sim p}\left[\frac{1}{\pi(\tau^*)}\right].$$

Note that this regret is a function of a particular policy $\pi(a \mid s)$, evaluated over potentially infinitely many steps in the original MDP. A policy that never replicates the target trajectory incurs infinite regret. Thus, we expect that optimal policies for the meta-POMDP will be stochastic.

## 4.2 SOLVING THE META-POMDP

We solve the meta-POMDP by finding an optimal distribution over trajectories:

$$\min_{\pi} \text{Regret}_p(\pi) \quad \text{s.t.} \quad \int \pi(\tau)d\tau = 1, \ \pi(\tau) > 0.$$

Using Lagrange multipliers (see Appendix A.1), we find that the optimal policy is:

$$\pi(\tau) = \frac{\sqrt{p(\tau)}}{\int \sqrt{p(\tau')}d\tau'}.$$

This policy is stochastic and matches the unnormalized distribution $\sqrt{p(\tau)}$. This result suggests that we can find the optimal policy by solving a MaxEnt RL problem, with a *trajectory-level* reward function $r_\tau(\tau) \triangleq \frac{1}{2}\log p(\tau)$. To make this statement precise, we consider the bandit setting and MDP setting separately. If the underlying MDP is a bandit, then trajectories are equivalent to actions. We can define a reward function as $r(a) = \frac{1}{2}\log p(a)$. Applying MaxEnt RL to this reward function yields the following policy, which is optimal for the meta-POMDP: $\pi(a) \propto \sqrt{p(a)}$. For MDPs with horizon lengths greater than one, we can make a similar statement:

**Lemma 4.1.** *Let a goal trajectory distribution $p(\tau)$ be given, and assume that there exists a policy $\pi$ whose trajectory distribution is proportional to the square-root of the target distribution: $\pi(\tau) \propto \sqrt{p(\tau)}$. Then there exists a reward function $r(s, a)$ such that the MaxEnt RL problem with $r(s, a)$ and the meta-POMDP have the same solution(s).*

*Proof.* Let $\pi^*$ be the solution to the meta-POMDP, so it must satisfy $\pi^*(\tau) \propto \sqrt{p(\tau)}$. Thus, $\pi^*$ is the solution to the MaxEnt RL problem with the trajectory-level reward $r(\tau) = \frac{1}{2}\log p(\tau)$:

$$\pi^* \in \arg\min_\pi \mathbb{E}_\pi \left[ \frac{1}{2}\log p(\tau) \right] + \mathcal{H}_\pi[a \mid s] = D_{\text{KL}}(\pi(\tau) \parallel \frac{1}{c}\sqrt{p(\tau)}) \iff \pi(\tau) \propto \sqrt{p(\tau)} \quad \forall \tau.$$

The normalizing constant $c$, which is independent from $\pi^*$, is introduced to handle the fact that $\sqrt{p(\tau)}$ does not integrate to one. The implication comes from the fact that the KL is minimized when its arguments are equal. We show in Appendix A.3 that a trajectory-level reward $r(\tau)$ can always be decomposed into an state-action reward $r(s_t, a_t)$ with the same MaxEnt RL solution. Thus, there exists a reward function such that MaxEnt RL solves the meta-POMDP:

$$\pi^* \in \arg\min_\pi \mathbb{E}_\pi \left[ \sum_{t=1}^T r(s_t, a_t) \right] + \mathcal{H}_\pi[a \mid s] \iff \pi^*(\tau) \propto \sqrt{p(\tau)} \quad \forall \tau. \qquad \square$$

The obvious criticism of the proof above is that it is not constructive, failing to specify how the MaxEnt RL reward might be obtained. Nonetheless, our analysis illustrates why MaxEnt RL methods might work well: even when the meta-POMDP is unknown, MaxEnt RL methods will minimize regret in *some* meta-POMDP, which could account for their good performance, particularly in the presence of uncertainty and perturbations.

### 4.3 GOAL-REACHING META-POMDPS

We can make the connection between MaxEnt RL and meta-POMDPs more precise by considering a special class of meta-POMDPs: meta-POMDPs where the target distribution is defined only in terms of the last state in a trajectory, corresponding to *goal-reaching* problems. While prior work on goal-reaching (Kaelbling, 1993; Schaul et al., 2015; Andrychowicz et al., 2017) assumes that the goal state is observed, the goal-reaching meta-POMDP only assumes that the policy has a belief about the goal state.

**Lemma 4.2.** *Let a meta-POMDP with target distribution that depends solely on the last state and action in the trajectory be given. That is, the target distribution $p(\tau)$ satisfies*

$$s_T(\tau) = s_T(\tau') \text{ and } a_T(\tau) = a_T(\tau') \implies p(\tau) = p(\tau') \, \forall \tau, \tau'$$

*where $s_T(\tau)$ and $a_T(\tau)$ are functions that extract the last state and action in trajectory $\tau$. We can thus write the density of a trajectory under the goal trajectory distribution as a function of the last state and action: $p(\tau) = \tilde{p}(s_T(\tau), a_T(\tau))$, where $\tilde{p}(s_T, a_T)$ is an unnormalized density. Assume that there exists a policy whose marginal state density at the last time step, $\rho_\pi^T(s, a)$, satisfies $\rho_\pi^T(s, a) \propto \sqrt{\tilde{p}(s, a)}$ for all states $s$. Then the MaxEnt RL problem with reward $r(s_t, a_t) \triangleq \frac{1}{2}\mathbb{1}(t = T) \cdot \log \tilde{p}(s_t, a_t)$ and the meta-POMDP have the same solutions.*

*Proof.* We simply combine Lemma 4.1 with the definition of the reward function $r$:

$$\sum_{t=1}^{T} r(s_t, a_t) = \sum_{t=1}^{T} \frac{1}{2} \mathbb{1}(t=T) \cdot \log p(s_t, a_t) = \frac{1}{2} \log \tilde{p}(s_T, s_T) = r_\tau(\tau). \qquad \square$$

While our assumption that there exists a policy that exactly matches some distribution ($\sqrt{\tilde{p}(s_T, a_T)}$) may seem somewhat unnatural, we provide a sufficient condition in Appendix A.4. Further, while the analysis so far has considered the equivalence of MaxEnt RL and the meta-POMDP at optimum, in Appendix A.5 we bound the difference between these problems away from their optima. In summary, the meta-POMDP allows us to represent goal-reaching tasks with uncertainty in the true goal state. Moreover, solving these goal-reaching meta-POMDPs with MaxEnt RL is straightforward, as the reward function for MaxEnt RL is a simple function of the last transition.

## 4.4 A COMPUTATIONAL EXPERIMENT

To conclude this section, we present a simple computational experiment to verify that Max-Ent RL does solve the meta-POMDP. We instantiated the meta-POMDP using 5-armed bandits. Each meta-POMDP is specified by a prior belief $p(i)$ over the target arm. We sample this distribution from a Dirichlet(1). To solve this meta-POMDP, we applied MaxEnt RL using a reward function $r(i) = \frac{1}{2} \log p(i)$, as derived above. When the agent pulls arm $i$, it observes a noisy reward $r_i \sim \mathcal{N}(r(i), 1)$. To implement the MaxEnt RL approach, we

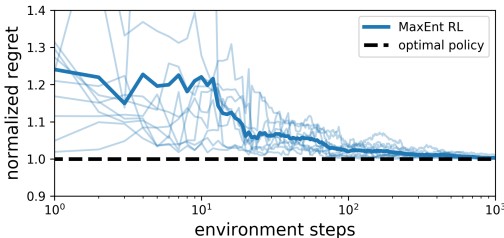

maintained the posterior of the reward $r(i)$, given our observations so far. We initialized our beliefs with a zero-mean, unit-variance Gaussian prior. To obtain a policy with MaxEnt RL, we chose actions with probability proportional to the exponentiated expected reward. Throughout training, we tracked the regret (Eq. 4.1). Since the minimum regret possible for each meta-POMDP is different, we normalized the regret for each meta-POMDP by dividing by the minimum possible regret. Thus, the normalized regret lies in the interval $[1, \infty]$, with lower being better. Figure 1 shows that MaxEnt RL converges to the regret-minimizing policy for each meta-POMDP. Code to reproduce all experiments is available online.[1]

Figure 1: **MaxEnt RL solves the Meta-POMDP.** On a 5-armed bandit problem, we show that solving a MaxEnt RL problem with a reward of $r(i) = \frac{1}{2} \log p(i)$ minimizes regret on the meta-POMDP defined by $p(i)$. The thick line is the average across 10 randomly-generated bandit problems (thin lines). Lower is better.

## 5 MAXIMUM ENTROPY RL AND ADVERSARIAL GAMES

While the meta-POMDP considered in the previous setting was defined in terms of task uncertainty, that uncertainty was fixed throughout the learning process. We now consider uncertainty introduced by an adversary who perturbs the reward function, and show how MaxEnt RL's aversion to deterministic policies provides robustness against these sorts of adversaries. In particular, we show MaxEnt RL is equivalent to solving *robust-reward control*, and run a computational experiment to support our claims. We generalize these results in Appendix B.

### 5.1 MAXENT RL SOLVES ROBUST-REWARD CONTROL

Our main result on reward robustness builds on the general equivalence between entropy maximization and game theory from prior work. To start, we note two results from prior work that show how entropy maximization can be written as a robust optimization problem:

**Lemma 5.1** (Grünwald et al. (2004)). *Let $x$ be a random variable, and let $\mathcal{P}$ be the set of all distributions over $x$. The problem of choosing a maximum entropy distribution for $x$ and maximizing the worst-case log-loss are equivalent:*

$$\max_{p \in \mathcal{P}} \mathcal{H}_p[x] = \max_{p \in \mathcal{P}} \min_{q \in \mathcal{P}} \mathbb{E}_p[-\log q(x)].$$

---

[1]Code: `https://drive.google.com/file/d/1Xf3OTxWBg67L2ka1eLd32qDQmtnG8Fxc`

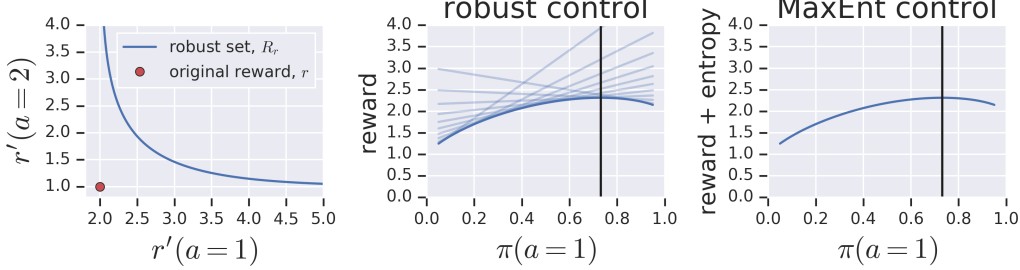

Figure 2: **MaxEnt RL = Robust-Reward Control**: The policy obtained by running MaxEnt RL on reward function $r$ is the optimal robust policy for a collection of rewards, $R_r$. *(Left)* We plot the original reward function, $r$ as a red dot, and the collection of reward functions, $R_r$, as a blue line. *(Center)* For each policy, parameterized solely by its probability of choosing action 1, we plot the expected reward for each reward function in $R_r$. The robust-reward control problem is to choose the policy whose worst-case reward (dark blue line) is largest. *(Right)* For each policy, we plot the MaxEnt RL objective (i.e., the sum of expected reward and entropy).

An immediately corollary is that maximizing the entropy of any *conditional* distribution is equivalent to a robust optimization problem:

**Corollary 5.1.1** (Grünwald et al. (2004); Ziebart et al. (2011)). *Let $x$ and $y$ be random variables, and let $\mathcal{P}_{x|y}$ be the set of all conditional distributions $p(x \mid y)$. The problem of choosing a maximum entropy distribution for the conditional distribution $p(x \mid y)$ and maximizing the worst-case log-loss of $x$ given $y$ are equivalent:*

$$\max_{p \in \mathcal{P}_{x|y}} \mathbb{E}_y \left[ \mathcal{H}_p[x \mid y] \right] = \max_{p \in \mathcal{P}_{x|y}} \min_{q \in \mathcal{P}_{x|y}} \mathbb{E}_y \left[ \mathbb{E}_p[-\log q(x \mid y)] \right].$$

In short, prior work shows that the principle of maximum entropy results minimizes worst-case performance on *prediction* problems that use *log-loss*. Our contribution extends this result to show that MaxEnt RL minimizes worst-case performance on *reinforcement learning* problems for certain classes of reward functions.

**Theorem 5.2.** *The MaxEnt RL objective for a reward function $r$ is equivalent to the robust-reward control objective for a certain class of reward functions:*

$$\mathbb{E}_\pi \left[ \sum_{t=1}^{T} r(s_t, a_t) \right] + \mathcal{H}_\pi[a \mid s] = \min_{r \in R_r} \mathbb{E}_\pi \left[ \sum_{t=1}^{T} r(s_t, a_t) \right],$$

*where*

$$R_r = \{ r'(s, a) \triangleq r(s, a) - \log q(a \mid s) \mid q \in \Pi \}. \tag{2}$$

For completeness, we provide a proof in Appendix B.1. We will call the set $R_r$ of reward functions a **robust set**. While the adversary in Theorem 5.2 may seem peculiar, this result can also be viewed as providing a lower bound on worst-case performance against an unknown reward function. As an aside, we note that the $\log q(a \mid s)$ term in the definition of the robust set arises from the fact that we consider MaxEnt RL algorithms using Shannon entropy. MaxEnt RL algorithms using other notions of entropy (Lee et al., 2019a; Chow et al., 2018) would result in different robust sets (see Grünwald et al. (2004)). We leave this generalization for future work.

## 5.2 A SIMPLE EXAMPLE

In Figure 2, we consider a simple, 2-armed bandit, with the following reward function:

$$r(a) = \begin{cases} 2 & a = 1 \\ 1 & a = 2 \end{cases}.$$

The robust set is then defined as

$$R_r = \left\{ r'(a) = \begin{cases} 2 - \log q(a) & a = 1 \\ 1 - \log(1 - q(a)) & a = 2 \end{cases} \,\middle|\, q \in [0, 1] \right\}.$$

Figure 2 (left) traces the original reward function and this robust set. Plotting the robust-reward control objective (center) and the MaxEnt RL objective (right), we observe that they are equivalent.

## 5.3 A COMPUTATIONAL EXPERIMENT

We ran an experiment to support our claim that MaxEnt RL is equivalent to solving the robust-reward problem. The mean for arm $i$, $\mu_i$, is drawn from a zero-mean, unit-variance Gaussian distribution, $\mu_i \sim \mathcal{N}(0, 1)$. When the agent pulled arm $i$, it observes a noisy reward $r_i \sim \mathcal{N}(\mu_i, 1)$. We implement the MaxEnt RL approaches as in Section 4.4. As a baseline, we compare to fictitious play (Brown, 1951), an algorithm for solving two-player, zero-sum games. Fictitious play alternates between choosing the best policy w.r.t. the historical average of observes rewards, and choosing the worst reward function for the historical average of policies. We choose the worst reward function from the robust set (Eq. 2). For fair comparison, the policy only observes the (noisy) reward associated with the selected arm. We also compared to an

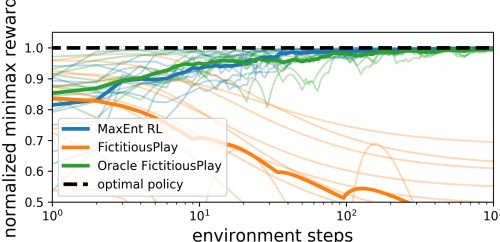

Figure 3: **MaxEnt RL solves a robust-reward control problem.** On a 5-armed bandit problem, MaxEnt RL converges to the optimal minimax policy. Fictitious play, a prior method for solving adversarial problems, fails to solve this task, but an oracle variant achieves reward similar to MaxEnt RL. The thick line is the average over 10 random seeds (thin lines). Higher is better.

oracle version of fictitious play that observes the (noisy) rewards associated with all arms, including arms not selected. We ran each method on the same set of 10 bandit problems, and evaluated the worst-case reward for each method (i.e., the expected reward of the policy, if the reward function were adversarially chosen from the robust set). Because each problem had a different minimax reward, we normalized the worst-case reward by the worst-case reward of the optimal policy. The normalized rewards are therefore in the interval $[0, 1]$, with 1 being optimal. Figure 3 plots the normalized reward throughout training. The main result is that MaxEnt RL converges to a policy that achieves optimal minimax reward, supporting our claim that MaxEnt RL is equivalent to a robust-reward control problem. The failure of fictitious play to solve this problem illustrates that the robust-reward control problem is not trivial to solve. Only the oracle version of fictitious play, which makes assumptions not made by MaxEnt RL, is competitive with MaxEnt RL. In Appendix C, we run a similar experiment on four robotic control tasks and find that MaxEnt RL optimizes the minimax reward better than standard RL and fictitious play.

## 6 DISCUSSION

In summary, this paper studies connections between MaxEnt RL and control problems with variability in the reward function. While MaxEnt RL is a relatively simple algorithm, the problems that it solves, such as robust-reward control and regret minimization in the meta-POMDP, are typically viewed as quite complex. This result hints that MaxEnt RL might also be used to solve even broader classes of control problems. The principle of maximum entropy has also been applied to inverse RL, and we encourage future work to consider whether MaxEnt IRL (Ziebart et al., 2008) is implicitly robust to some sort of variability. Finally, we speculate that our results may help understand behavior in the natural world. The abundance of evidence for probability matching in nature suggests that, in the course of evolution, creatures that better handled uncertainty and avoided adversaries were more likely to survive. We encourage RL researchers to likewise focus their research on problem settings likely to occur in the real world.

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

## A  META-POMDP

### A.1  SOLVING FOR THE OPTIMAL TRAJECTORY DISTRIBUTION

We will solve the optimization problem introduced in Section 4.2 using the constrained Euler-Lagrangian method. The Lagrangian is

$$\mathcal{L}(\pi, \lambda) \triangleq \int \frac{p(\tau)}{\pi(\tau)} d\tau + \lambda \left( \int \pi(\tau) d\tau - 1 \right).$$

The first and second derivatives of the Lagrangian are:

$$\frac{d\mathcal{L}}{d\pi(\tau)} = -\frac{p(\tau)}{\pi(\tau)^2} - \lambda \qquad \frac{d^2\mathcal{L}}{d\pi^2} = 2\frac{p(\tau)}{\pi(\tau)^3} > 0.$$

Note that the second derivative is positive so setting the first derivative equal to zero will provide a minimum of the objective:

$$\frac{d\mathcal{L}}{d\pi(\tau)} = 0 \implies \pi(\tau) = \sqrt{-\lambda p(\tau)}.$$

We then solve for $\lambda$ using the constraint that $\pi(\tau)$ integrate to one, yielding the solution to the optimization problem:

$$\pi(\tau) = \frac{\sqrt{p(\tau)}}{\int \sqrt{p(\tau')}d\tau'}.$$

## A.2 The Meta-POMDP as (Memoryless) Meta-Learning

The meta-POMDP can be viewed as a meta-learning problem (Thrun & Pratt, 2012), solved with a memoryless meta-learner. Formally, a meta-learning algorithm is a distribution over policies $\pi$, given the observed data, $\mathcal{D}$: $p(\pi \mid \mathcal{D})$. We will consider memoryless meta-learning algorithms: the distribution of policies proposed at each step is the same, as the meta-learner cannot update its beliefs based on observed evidence: $p(\pi \mid \mathcal{D}) = p(\pi)$. We define a meta-learning problem by a distribution over MDPs, $p(\mathcal{M})$. A unknown MDP $\mathcal{M}$ will be sampled, and the job of the meta-learning algorithm is to solve $\mathcal{M}$ as quickly as possible. Solving an MDP can mean a number of different things: reaching a goal state, achieving a certain level of reward, or avoiding episode termination. For simplicity, we assume that each MDP has a single successful policy, and each policy is successful for a single MDP, though this analysis can likely be extended to more general notions of success. We will define $\pi_i$ as the unique policy that solves MDP $\mathcal{M}_i$. Note that, in this setting, the regret of the meta-POMDP is exactly the number of episodes required to find the optimal policy for the (unknown) MDP. Lemma 4.1 tells us that the optimal (memoryless) meta-learning algorithm is defined as $p(\pi_i) \propto \sqrt{p(\mathcal{M}_i)}$.

While the assumption that the meta-learning algorithm is independent of observed data is admittedly strong, it is realistic in settings where failing at one task provides no information about that the true task might be. Broadly, we believe that the meta-POMDP is a first step towards understanding if and how MaxEnt RL might be used to solve problems typically approached as meta-learning problems.

## A.3 When Can Trajectory-Level Rewards be Decomposed?

**Lemma A.1.** *Let a trajectory-level reward function $r_\tau(\tau)$ be given, and define the corresponding target distribution as*

$$p_{r_\tau}(\tau) \propto p_1(s_1)e^{r_\tau(\tau)}\prod_{t=1}^{T}p(s_{t+1} \mid s_t, a_t).$$

*If there exists a Markovian policy $\pi$ such that $\pi(\tau) = p_{r_\tau}(\tau)$ for all trajectories $\tau$, then there exists a state-action level reward function $r(s, a)$ satisfying*

$$r_\tau(\tau) = \sum_{t=1}^{T}r(s_t, a_t) \qquad \forall\tau,$$

*Proof.* To start, we recall the definitions of $\pi(\tau)$ and $p_{r_\tau}(\tau)$:

$$\pi(\tau) \propto p_1(s_1)\prod_{t=1}^{T}p(s_{t+1} \mid s_t, a_t)\pi(a_t \mid s_t) \tag{3}$$

$$p_{r_\tau}(\tau) \propto p_1(s_1)e^{r_\tau(\tau)}\prod_{t=1}^{T}p(s_{t+1} \mid s_t, a_t). \tag{4}$$

By our assumption that $\pi(\tau) = p_{r_\tau}(\tau)$, we know that Equations 3 and 4 are equal, up to some proportionality constant $c'$:

$$p_1(s_1)e^{r_\tau(\tau)} \prod_{t=1}^{T} p(s_{t+1} \mid s_t, a_t) = c' p_1(s_1) \prod_{t=1}^{T} p(s_{t+1} \mid s_t, a_t)\pi(a_t \mid s_t)$$

$$= p_1(s_1)e^{\log c' + \sum_{t=1}^{T} \log \pi(a_t|s_t)} \prod_{t=1}^{T} p(s_{t+1} \mid s_t, a_t)$$

$$= p_1(s_1)e^{\sum_{t=1}^{T} r(s_t, a_t)} \prod_{t=1}^{T} p(s_{t+1} \mid s_t, a_t),$$

where $r(s, a) \triangleq \log \pi(a \mid s) + \frac{1}{T}\log c'$. $\qquad\square$

## A.4 WHEN DOES A SOLUTION EXIST?

When considering goal-reaching meta-POMDPs, we made an assumption that there exists a policy that exactly matches some distribution ($\sqrt{\tilde{p}(s_T, a_T)}$). Here, we provide a sufficient (but not necessary) condition for the existence of such a policy

**Lemma A.2.** *Let $p(\tau) = \tilde{p}(s_T, a_T)$ be some distribution over trajectories that depends only on the last state and action in each trajectory (as in Lemma 4.2). If, for every state $s_T$ and action $a_T$ where $\tilde{p}(s_T, a_T) > 0$, there exists a policy $\pi_{s_T, a_T}$ that deterministically reaches state $s_T$ and action $a_T$ on the final time step, then there exists a Markovian policy satisfying $\pi(\tau) = p(\tau)$.*

The main idea behind the proof is that, if there exists policies that reach each of the possible target state-action pairs, then there exists a way of "mixing" these policies to obtain a policy with the desired marginal distribution.

*Proof.* First, we construct a mixture policy $\bar{\pi}$ by sampling $s_T, a_T \sim \tilde{q}(s_T, a_T)$ at the start of each episode, and using policy $\pi_{s_T, a_T}$ for every step in that episode. By construction, we have $\rho_{\bar{\pi}}(s_T, a_T) = q(s_T, a_T)$. However, this policy $\bar{\pi}$ is non-Markovian. Nonetheless, Ziebart (2010, Theorem 2.8) guarantees that there exists a Markovian policy $\tilde{\pi}$ with the same marginal state distribution: $\tilde{\pi}(s_T, a_T) = \bar{\pi}(s_T, a_T)$. Thus, there exists a Markovian policy, $\tilde{\pi}$ satisfying $\tilde{\pi}(s_T, a_T) = q(s_T, a_T)$. $\qquad\square$

## A.5 BOUNDING THE DIFFERENCE BETWEEN MAXENT RL AND THE META-POMDP

While the result in Section 4 shows that MaxEnt RL has the same solution as the meta-POMDP, it does not tells us how the two control problems differ away from their optima. The following theorem provides an answer.

**Theorem A.3.** *Assume that the ratio $p(\tau)/\pi(\tau)$ is bounded in $[a, b]$ for all trajectories $\tau$ and $a, b > 0$. Further, assume that there exists a policy $\pi_r$ that can solve the MaxEnt RL problem exactly (i.e., $\pi(\tau) = p(\tau)$). Then the MaxEnt RL objective minimizes an upper bound on the log regret of the meta-POMDP, plus an additional term that vanishes as $\pi \to p_r$:*

$$J(\pi, p_r) \geq \log Regret_{p_r^2}(\pi) + C(\pi, r).$$

Before proving this theorem, we note that the assumption that the MaxEnt RL problem can be solved exactly is always satisfied for linearly-solveable MDPs (Todorov, 2007). Moreover, given a MDP that cannot be solved exactly, we can always modify the reward function (i.e., the target distribution $p(\tau)$) such that the optimal policy remains the same, but such that the optimal policy now exactly matches the target distribution. The proof of the theorem will consist of two steps. First, we will bound the difference between the log regret of the meta-POMDP and a *forward KL*. The second step is to bound the difference between that forward KL and the reverse KL (which is optimized by MaxEnt RL).

*Proof.* To start, we apply a "backwards" version of Jensen's inequality from Simic (2009, Theorem 1.1), which states that the following inequality holds for any convex function $f(x)$:

$$\int p(\tau)f(x_\tau)d\tau - f\int p(\tau)x_\tau d\tau \leq \frac{1}{4}(b - a)(f'(b) - f'(a)).$$

We use $-\log(x)$ as our convex function, whose derivative is $\frac{-1}{x}$, and further define $x_\tau = p(\tau)/\pi(\tau)$:

$$-\int p(\tau) \log \left( \frac{p(\tau)}{\pi(\tau)} \right) d\tau + \log \int \frac{p^2(\tau)}{\pi(\tau)} d\tau \leq \frac{1}{4}(b-a)(1/a - 1/b).$$

Rearranging terms, we get

$$\log \int \frac{p^2(\tau)}{\pi(\tau)} d\tau \leq \int p(\tau) \log \left( \frac{p(\tau)}{\pi(\tau)} \right) d\tau + \frac{1}{4}(b-a)(1/a - 1/b). \tag{5}$$

The LHS is not quite a log regret (Eq. 4.1) because it contains a $p^2(\tau)$ term in the numerator, which is not a proper probability distribution. Defining $Z = \int p^2(\tau)$ as the normalizing constant (which does not depend on $\pi$), we can write the log regret using a proper distribution:

$$\log \mathrm{Regret}_{\frac{1}{Z} p^2}(\pi) = \log \int \frac{p^2(\tau)}{Z\pi(\tau)} d\tau = \log \int \frac{p^2(\tau)}{\pi(\tau)} d\tau - \log Z$$

We can now rewrite Equation 5 as a bound on log regret:

$$\log \mathrm{Regret}_{\frac{1}{Z} p^2}(\pi) \leq \int p(\tau) \log \left( \frac{p(\tau)}{\pi(\tau)} \right) d\tau + \frac{1}{4}(b-a)(1/a - 1/b) + \log Z. \tag{6}$$

Notice that the integral on the RHS is the forward KL between $p$ and $\pi$, whereas MaxEnt RL minimizes the reverse KL. Our next step is to show that the forward KL is not too much larger than the reverse KL, so optimizing the reverse KL (as done by MaxEnt RL) will still minimize an upper bound on the log regret of the meta-POMDP. We will do this using a result from Norouzi et al. (2016). First, we need to define the logits corresponding to distributions $\pi$ and $p$. We start by defining the logits for just the dynamics:

$$\ell_d(\tau) \triangleq \log p_1(s_1) + \sum_{t=1}^{T} \log p(s_{t+1} \mid s_t, a_t).$$

Now, the policy distribution $\pi(\tau)$ and the target distribution, $p(\tau)$, can both be written in terms of dynamics and policy:

$$\log \pi(\tau) = \ell_d(\tau) + \sum_{t=1}^{T} \log \pi(a_t \mid s_t)$$

$$\log p(\tau) = \ell_d(\tau) + \sum_{t=1}^{T} \log \pi_p^*(a_t \mid s_t),$$

where $\pi_p^*$ is the optimal MaxEnt RL policy for the reward $r(\tau) = \log p(\tau)$. By our assumption that there exists some $\pi$ such that $\pi(\tau) = p(\tau)$, we know that $\pi_p^*(\tau) = p(\tau)$. With this notation in place, we can employ Proposition 2 of Norouzi et al. (2016):

$$D_{\mathrm{KL}}(p(\tau) \parallel \pi(\tau)) \leq D_{\mathrm{KL}}(\pi(\tau) \parallel p(\tau)) + \sum_\tau (\ell_p(\tau) - \ell_\pi(\tau))^2$$

$$= D_{\mathrm{KL}}(\pi(\tau) \parallel p(\tau)) + \sum_\tau \left( \sum_{t=1}^{T} \log \pi_p^*(a_t \mid s_t) - \sum_{t=1}^{T} \log \pi(a_t \mid s_t) \right)^2. \tag{7}$$

Note that the dynamics logits $\ell_d$ cancelled with one another. Now, we combine Equations 6 and 7 to obtain:

$$\log \mathrm{Regret}_{p_r^2}(\pi) \leq \frac{1}{Z} D_{\mathrm{KL}}(\pi(\tau) \parallel p(\tau)) + C(\pi, r), \tag{8}$$

where

$$C(\pi, r) = \left( \sum_{t=1}^{T} \log \pi_p^*(a_t \mid s_t) - \sum_{t=1}^{T} \log \pi(a_t \mid s_t) \right)^2 + \frac{1}{4}(b-a)(1/a - 1/b).$$

As $\pi \to \pi_p^*$, difference of log probabilities (term 1) vanishes. Additionally, the ratio $p(\tau)/\pi(\tau) \to 1$, so we can take $a$ and $b$ (the limits on the probability ratio) towards 1. As $a \to 1$ and $b \to 1$, the second term in $C(\pi, r)$ vanishes as well. $\qquad \square$

In summary, the difference between the MaxEnt RL objective and the log regret of the meta-POMDP is controlled by a term $C(\pi, r)$. At the solution to the MaxEnt RL problem, this term is zero, implying that solving the MaxEnt RL problem will minimize regret on the meta-POMDP.

# B  MAXIMUM ENTROPY RL AND ADVERSARIAL GAMES

## B.1  PROOF AND EXTENSIONS OF THEOREM 5.2

Our proof of Theorem 5.2 consists of first applying Corollary 5.1.1, and then rearranging terms.

*Proof.*

$$\max_{\pi \in \Pi} \mathbb{E}_{\pi} \left[ \sum_{t=1}^{T} r(s_t, a_t) \right] + \mathcal{H}_{\pi}[a \mid s] = \max_{\pi \in \Pi} \mathbb{E}_{\pi} \left[ \sum_{t=1}^{T} r(s_t, a_t) \right] + \min_{q \in \Pi} \mathbb{E}_{\pi} \left[ \sum_{t=1}^{T} - \log q(a_t \mid s_t) \right]$$

$$= \max_{\pi \in \Pi} \min_{q \in \Pi} \mathbb{E}_{\pi} \left[ \sum_{t=1}^{T} \left( r(s_t, a_t) - \log q(a_t \mid s_t) \right) \right]$$

$$= \max_{\pi \in \Pi} \min_{r' \in R_r} \mathbb{E}_{\pi} \left[ \sum_{t=1}^{T} r'(s_t, a_t) \right].$$

$\square$

A corollary of this result is that the solution to the MaxEnt RL objective is robust. More precisely, the MaxEnt objective obtained by some policy is a lower bound on that policy's reward for *any* reward function in some set.

**Corollary B.0.1.** *Let policy $\pi$ and reward function $r$ be given, and let $J$ be the MaxEnt objective policy $\pi$ on reward function $r$:*

$$J(\pi, r) \triangleq \mathbb{E}_{\pi} \left[ \sum_{t=1}^{T} r(s_t, a_t) \right] + \mathcal{H}_{\pi}[a \mid s].$$

*Then the expected return of policy $\pi$ on any reward function in $R_r$ is at least $J$:*

$$\mathbb{E}_{\pi} \left[ \sum_{t=1}^{T} r'(s_t, a_t) \right] \geq J(\pi, r) \quad \forall r' \in R_r.$$

## B.2  INTUITION FOR ROBUST SETS

We can gain intuition for the robust set by explicitly writing out the definition of $\Pi$:

$$\Pi \triangleq \left\{ q \mid q(s, a) \in [0, 1] \; \forall s, a \quad \text{and} \quad \int_A q(a \mid s) da = 1 \; \forall s \right\}$$

$$= \left\{ q \mid \log q(s, a) \in [-\infty, 0] \; \forall s, a \quad \text{and} \quad \int_A e^{\log q(a \mid s)} da = 1 \; \forall s \right\}.$$

Now, we can rewrite our definition of $R_r$ as follows:

$$R_r = \left\{ r(s, a) + u_s(a) \mid u_s(a) \geq 0 \; \forall s, a \text{ and } \int_{\mathcal{A}} e^{-u_s(a)} da = 1 \; \forall s \right\} \tag{9}$$

$$= \left\{ r'(s, a) \mid r'(s, a) \geq 0 \; \forall s, a \text{ and } \int_{\mathcal{A}} e^{r(s,a) - r'(s,a)} da = 1 \; \forall s \right\}.$$

Intuitively, we are robust to all reward functions obtained by adding (positive) additional reward to the original reward, with the only constraints being that (1) the reward function is increased enough, and (2) significantly increasing the reward for one state-action pair limits the amount the reward can be increased for another state-action pair. One important caveat of the results in this section is that, individually, the reward functions in in the robust set $R_r$ are "easier" than the original reward function, in that they assign larger values to the reward at a given state and action:

$$r(s, a) \leq r'(s, a) \quad \forall s \in \mathcal{S}, a \in \mathcal{A}, r' \in R_r.$$

Appendix B.4 discusses the role of temperatures on the robustness of MaxEnt RL.

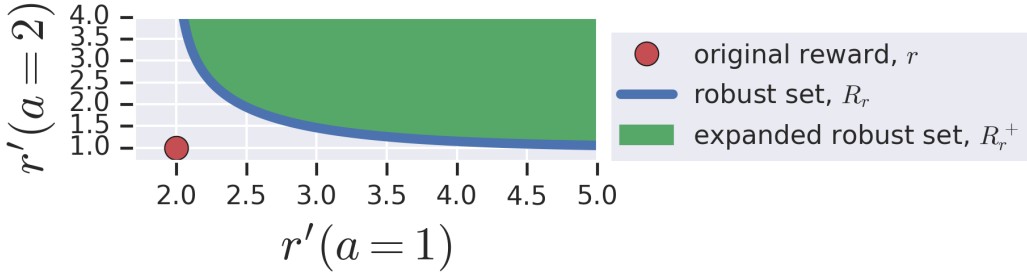

Figure 4: **Expanded Robust Set**: MaxEnt RL is robust to the reward functions along the blue line, which implies that it is also robust to the reward functions in the shaded region.

### B.3 EQUIVALENCE CLASSES OF ROBUST REWARD CONTROL PROBLEMS

In this section, we aim to understand whether the policy obtained by running MaxEnt RL on reward function $r$ is robust to reward functions besides those in $R_r$. As a first step, we show that the policy is also robust to reward functions that are "easier" than those in $R_r$. We then show that the set $R_r$ is not unique and introduce a family of equivalent robust-reward control problems, all of which are equivalent to the original MaxEnt RL problem.

#### B.3.1 ROBUSTNESS TO DOMINATED REWARD FUNCTIONS

In Theorem 5.2, we showed that the optimal MaxEnt RL policy $\pi$ is also the minimax policy for the reward functions in $R_r$. A somewhat trivial corollary is that $\pi$ is also robust to any reward function that is pointwise weakly better than a reward function in our robust set:

**Lemma B.1.** *The MaxEnt RL objective for a reward function $r$ is equivalent to the robust-reward control objective for a class of reward functions, $R_r^+ \supseteq R_r$:*

$$\mathbb{E}_\pi \left[ \sum_{t=1}^T r(s_t, a_t) \right] + \mathcal{H}_\pi[a \mid s] = \min_{r \in R_r^+} \mathbb{E}_\pi \left[ \sum_{t=1}^T r(s_t, a_t) \right],$$

*where*

$$R_r^+ \triangleq \{ r^+(s, a) = r'(s, a) + c \mid c \geq 0, r' \in R_r \}.$$

*Proof.* To prove this, we simply note that these additional reward functions, $r^+ \in R_r^+ \setminus R_r$, will never be chosen as the arg min of the RHS. Thus, expanding the constraint set from $R_r$ to $R_r^+$ does not change the value of the RHS. □

This constrained says that we are robust to reward functioned that are bounded away from the original reward function. We plot the expanded robust set, $R_r^+$, in Figure 4. Note that the expanded robust set corresponds to all reward functions "above" and to the "right" of our original robust set. We can use this result to write a new definition for the robust set. Since we now know that $u$ can be made arbitrarily small, we can allow $e^{-u_s(a)}$ to take very small values. More precisely, whereas before we were constrained to add a term that integrated to one, we now are allows to add any term whose integral is at at most one:

$$R_r^+ = \left\{ r(s, a) + u_s(a) \mid u_s(a) \geq 0 \; \forall s, a \text{ and } \int_{\mathcal{A}} e^{-u_s(a)} da \leq 1 \; \forall s \right\}.$$

#### B.3.2 GLOBAL AFFINE TRANSFORMATIONS

In optimal control, modifying a reward function by adding a global constant or scaling *all* reward by a positive constant does not change the optimal policy. Thus, the robust-reward control problem for a set of rewards $R_r$ has the same solution as the robust-reward control problem for a scaled and shifted set of rewards:

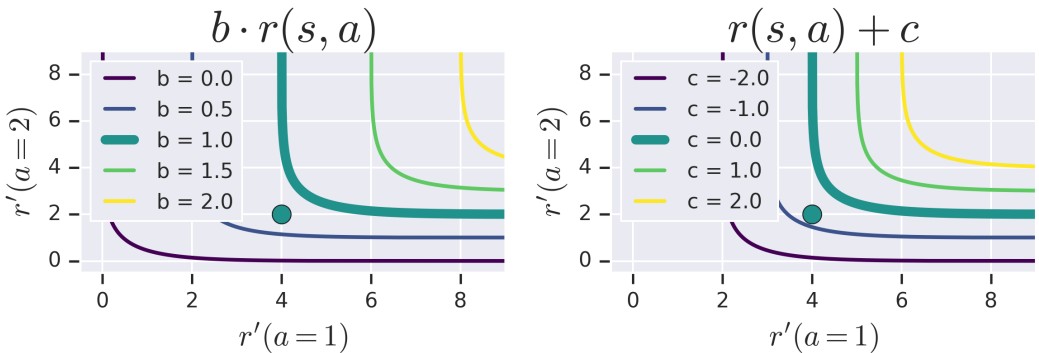

Figure 5: **Global Affine Transformations**: In a simple, 2-armed bandit, we draw one reward function (turquoise dot) an its robust set (turquoise, thick line). The policy that is minimax for the reward functions in the robust set. The policy is also minimax for other robust sets, obtained by *(Left)* scaling and *(Right)* shifting the original robust set. Importantly, the policy is not *simultaneously* robust against the *union* of these robust sets.

**Lemma B.2.** *Let a set of reward functions $R_r$ be given, and let $b, c \in R, b > 0$ be arbitrary constants. Then the following two optimization problems are equivalent:*

$$\arg\max_{\pi} \min_{r' \in R_r} \mathbb{E}_{\pi}\left[\sum_{t=1}^{T} r'(s_t, a_t)\right] = \arg\max_{\pi} \min_{r' \in bR_r + c} \mathbb{E}_{\pi}\left[\sum_{t=1}^{T} r'(s_t, a_t)\right].$$

*Proof.* The proof follows simply by linearity of expectation, and the invariance of the argmax to positive affine transformations:

$$\arg\max_{\pi} \min_{r' \in bR_r + c} \mathbb{E}_{\pi}\left[\sum_{t=1}^{T} r'(s_t, a_t)\right] = \arg\max_{\pi} \min_{r' \in R_r} \mathbb{E}_{\pi}\left[\sum_{t=1}^{T}(br'(s_t, a_t) + c)\right] \qquad (10)$$

$$= \arg\max_{\pi} b \min_{r' \in R_r} \mathbb{E}_{\pi}\left[\sum_{t=1}^{T} r'(s_t, a_t)\right] + cT \qquad (11)$$

$$= \arg\max_{\pi} \min_{r' \in R_r} \mathbb{E}_{\pi}\left[\sum_{t=1}^{T} r'(s_t, a_t)\right]. \qquad (12)$$

$\square$

Note, however, that the robust-reward control problem for rewards $R_r$ is not the same as being *simultaneously* robust to the union of all affine transformations of robust sets. Said another way, there exists a *family* of equivalent robust-reward control problems, each defined by a fixed affine transformation of $R_r$.

To gain some intuition for these transformations of reward functions, we apply a variety of transformations to the reward function from Figure 2. In Figure 5 (left) we show the effect of multiplying the reward by a positive constant. Figure 5 (right) shows the effect of adding a constant to the reward for every state and action.. For the robust sets in the right plot, there exists another reward function ($r^{\dagger} = r + c$) such that the shifted robust set is equal to the robust set of the shifted reward (i.e., $R_{r+c} = R_r + c$):

**Lemma B.3.** *Let reward function $r$ and constant $c \in R$ be given. Define a reward function $r_c(s, a) \triangleq r(s, a) + c$. Let $\mathcal{W}_r$ be the set of robust optimization problems which are equivalent to the MaxEnt RL problem on reward function $r$. The MaxEnt RL problem with the shifted reward function, $r_c$, is equivalent to the same set of robust optimization problems: $\mathcal{W}_r = \mathcal{W}_{r_c}$*

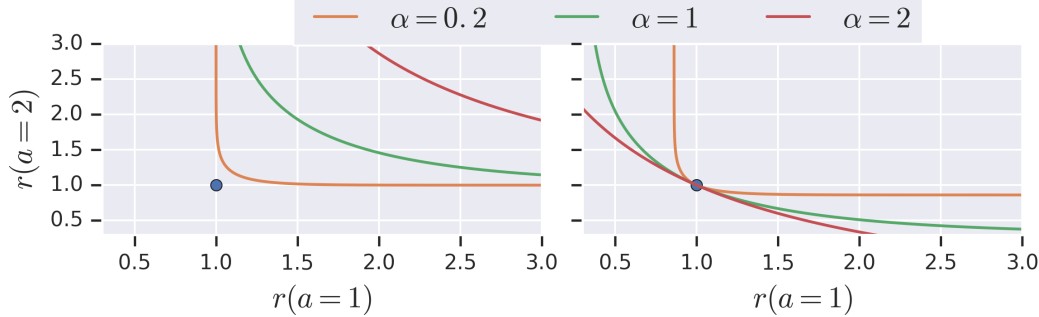

Figure 6: **Effect of Temperature**: *(Left)* For a given reward function (blue dot), we plot the robust sets for various values of the temperature. Somewhat surprisingly, it appears that increasing the temperature decreases the set of reward functions that MaxEnt is robust against. *(Right)* We examine the opposite: for a given reward function, which other robust sets might contain this reward function. We observe that robust sets corresponding to larger temperatures (i.e., the red curve) can be simultaneously robust against more reward functions than robust sets at lower temperatures.

*Proof.* In addition to the argument given above, we can simply note that the two MaxEnt RL problems are the same:

$$\arg\max_{\pi} \mathbb{E}_{\pi}\left[\sum_{t=1}^{T} r_c(s_t, a_t)\right] + \mathcal{H}_{\pi}[a \mid s] = \arg\max_{\pi} \mathbb{E}_{\pi}\left[\sum_{t=1}^{T} r(s_t, a_t)\right] + \mathcal{H}_{\pi}[a \mid s] + \overset{\text{const.}}{\cancel{T \cdot c.}}$$

$\square$

Lemma B.3 is not true for the robust sets in the left plot. While the policy that is minimax for $R_r$ is also minimax for $bR_r$, there does not exist another reward function $r^{\dagger}$ such that $R_{r^{\dagger}} = bR_r$. The reason is that scaling the robust violates the constraint $\int e^{-u_s(a)} da = 1$.

## B.4 TEMPERATURES

Many algorithms for MaxEnt RL (Haarnoja et al., 2018b; Fox et al., 2015; Nachum et al., 2017) include a temperature $\alpha > 0$ to balance the reward and entropy terms:

$$J(\pi, r) = \mathbb{E}_{\pi}\left[\sum_{t=1}^{T} r(s_t, a_t)\right] + \alpha \mathcal{H}_{\pi}[a \mid s].$$

We can gain some intuition into the effect of this temperature on the set of reward functions to which we are robust. In particular, including a temperature $\alpha$ results in the following robust set:

$$R_r^{\alpha} = \left\{ r(s, a) + \alpha u_s(a) \mid u_s(a) \geq 0 \ \forall s, a \text{ and } \int_{\mathcal{A}} e^{-u_s(a)} da \leq 1 \ \forall s \right\} \quad (13)$$

$$= \left\{ r(s, a) + u_s(a) \mid u_s(a) \geq 0 \ \forall s, a \text{ and } \int_{\mathcal{A}} e^{-u_s(a)/\alpha} da \leq 1 \ \forall s \right\}. \quad (14)$$

In the second line, we simply moved the temperature from the objective to the constraint by redefining $u_s(a) \to \frac{1}{\alpha} u_s(a)$.

We visualize the effect of the temperature in Figure 6. First, we fix a reward function $r$, and plot the robust set $R_r^{\alpha}$ for varying values of $\alpha$. Figure 6 (left) shows the somewhat surprising result that increasing the temperature (i.e., putting more weight on the entropy term) makes the policy *less* robust. In fact, the robust set for higher temperatures is a strict subset of the robust set for lower temperatures:

$$\alpha_1 < \alpha_2 \implies R_r^{\alpha_2} \subseteq R_r^{\alpha_2}.$$

This statement can be proven by simply noting that the function $e^{-\frac{x}{\alpha}}$ is an increasing function of $\alpha$ in Equation 14. It is important to recognize that being robust against more reward functions is not always desirable. In many cases, to be robust to everything, an optimal policy must do nothing.

We now analyze the temperature in terms of the converse question: if a reward function $r'$ is included in a robust set, what other reward functions are included in that robust set? To do this, we take a reward function $r'$, and find robust sets $R_r^\alpha$ that include $r'$, for varying values of $\alpha$. As shown in Figure 6 (right), if we must be robust to $r'$ and use a high temperature, the only other reward functions to which we are robust are those that are similar, or pointwise weakly better, than $r'$. In contrast, when using a small temperature, we are robust against a wide range of reward functions, including those that are highly dissimilar from our original reward function (i.e., have higher reward for some actions, lower reward for other actions). Intuitively, increasing the temperature allows us to simultaneously be robust to a larger set of reward functions.

### B.5 MaxEnt Solves Robust Control for Rewards

In Section 5, we showed that MaxEnt RL is equivalent to some robust-reward problem. The aim of this section is to go backwards: given a set of reward functions, can we formulate a MaxEnt RL problem such that the robust-reward problem and the MaxEnt RL problem have the same solution?

**Theorem B.4.** *For any collection of reward functions $R$, there exists another reward function $r$ such that the MaxEnt RL policy w.r.t. $r$ is an optimal robust-reward policy for $R$:*

$$\arg\max_\pi \mathbb{E}_\pi \left[ \sum_{t=1}^{T} r(s_t, a_t) \right] + \mathcal{H}_\pi[a \mid s] \subseteq \arg\max_\pi \min_{r' \in R} \mathbb{E}_\pi \left[ \sum_{t=1}^{T} r'(s_t, a_t) \right].$$

We use set containment, rather than equality, because there may be multiple solutions to the robust-reward control problem.

*Proof.* Let $\pi^*$ be a solution to the robust-reward control problem:

$$\pi^* \in \arg\max_\pi \min_{r_i \in R} \mathbb{E}_\pi \left[ \sum_{t=1}^{T} r_i(s_t, a_t) \right].$$

Define the MaxEnt RL reward function as follows:

$$r(s, a) = \log \pi^*(a \mid s).$$

Substituting this reward function in Equation 2, we see that the unique solution is $\pi = \pi^*$. □

Intuitively, this theorem states that we can use MaxEnt RL to solve *any* robust-reward control problem that requires robustness with respect to any arbitrary set of rewards, if we can find the right corresponding reward function $r$ for MaxEnt RL. One way of viewing this theorem is as providing an avenue to sidestep the challenges of robust-reward optimization. Unfortunately, we will still have to perform robust optimization to learn this magical reward function, but at least the cost of robust optimization might be amortized. In some sense, this result is similar to Ilyas et al. (2019).

### B.6 Finding the Robust Reward Function

In the previous section, we showed that a policy robust against any set of reward functions $R$ can be obtained by solving a MaxEnt RL problem. However, this requires calculating a reward function $r^*$ for MaxEnt RL, which is not in general an element in $R$. In this section, we aim to find the MaxEnt reward function that results in the optimal policy for the robust-reward control problem. Our main idea is to find a reward function $r^*$ such that its robust set, $R_{r^*}$, contains the set of reward functions we want to be robust against, $R$. That is, for each $r_i \in R$, we want

$$r_i(s, a) = r^*(s, a) + u_s(a) \quad \text{for some } u_s(a) \text{ satisfying} \quad \int_{\mathcal{A}} e^{-u_s(a)} da \leq 1 \; \forall s.$$

Replacing $u$ with $r' - r^*$, we see that the MaxEnt reward function $r$ must satisfy the following constraints:

$$\int_{\mathcal{A}} e^{r^*(s,a) - r'(s,a)} da \leq 1 \; \forall s \in \mathcal{S}, r' \in R.$$

We define $R^*(R)$ as the set of reward functions satisfying this constraint w.r.t. reward functions in $R$:

$$R^*(R) \triangleq \left\{ r^* \ \middle| \ \int_{\mathcal{A}} e^{r^*(s,a) - r'(s,a)} da \leq 1 \ \forall s \in \mathcal{S}, r' \in R \right\}$$

Note that we can satisfy the constraint by making $r^*$ arbitrarily negative, so the set $R^*(R)$ is non-empty. We now use Corollary 5.1.1 to argue that all any applying MaxEnt RL to any reward function in $r^* \in R^*(R)$ lower bounds the robust-reward control objective.

**Corollary B.4.1.** *Let a set of reward functions $R$ be given, and let $r^* \in R^*(R)$ be an arbitrary reward function belonging to the feasible set of MaxEnt reward functions. Then*

$$J(\pi, r^*) \leq \min_{r' \in R} \mathbb{E}_\pi \left[ \sum_{t=1}^T r'(s_t, a_t) \right] \qquad \forall \pi \in \Pi.$$

Note that this bound holds for all feasible reward functions and all policies, so it also holds for the maximum $r^*$:

$$\max_{r^* \in R^*(R)} J(\pi, r^*) \leq \min_{r' \in R} \mathbb{E}_\pi \left[ \sum_{t=1}^T r'(s_t, a_t) \right] \qquad \forall \pi \in \Pi. \tag{15}$$

Defining $\pi^* = \arg\max_\pi J(\pi, r^*)$, we get the following inequality:

$$\max_{r^* \in R^*(R), \pi \in \Pi} J(\pi, r^*) \leq \min_{r' \in R} \mathbb{E}_{\pi^*} \left[ \sum_{t=1}^T r'(s_t, a_t) \right] \leq \max_{\pi \in \Pi} \min_{r' \in R} \mathbb{E}_\pi \left[ \sum_{t=1}^T r'(s_t, a_t) \right]. \tag{16}$$

Thus, we can find the tightest lower bound by finding the policy $\pi$ and feasibly reward $r^*$ that maximize Equation 16:

$$\max_{r, \pi} \quad \mathbb{E}_\pi \left[ \sum_{t=1}^T r(s_t, a_t) \right] + \mathcal{H}_\pi[a \mid s] \tag{17}$$

$$\text{s.t.} \quad \int_{\mathcal{A}} e^{r(s,a) - r'(s,a)} da \leq 1 \ \forall \ s \in \mathcal{S}, r' \in R.$$

It is useful to note that the constraints are simply LogSumExp functions, which are convex. For continuous action spaces, we might approximate the constraint via sampling. Given a particular policy, the optimization problem w.r.t. $r$ has a linear objective and convex constraint, so it can be solved extremely quickly using a convex optimization toolbox. Moreover, note that the problem can be solved independently for every state. The optimization problem is not necessarily convex in $\pi$.

### B.7 ANOTHER COMPUTATIONAL EXPERIMENT

This section presents an experiment to study the approach outlined above. Of particular interest is whether the lower bound (Eq 16) comes close the the optimal minimax policy.

We will solve robust-reward control problems on 5-armed bandits, where the robust set is a collection of 5 reward functions, each is drawn from a zero-mean, unit-variance Gaussian. For each reward function, we add a constant to all of the rewards to make them all positive. Doing so guarantees that the optimal minimax reward is positive. Since different bandit problems have different optimal minimax rewards, we will normalize the minimax reward so the maximum possible value is 1.

Our approach, which we refer to as "LowerBound + MaxEnt", solves the optimization problem in Equation 17 by alternating between (1) solving a convex optimization problem to find the optimal reward function, and (2) computing the optimal MaxEnt RL policy for this reward function. Step 1 is done using CVXPY, while step 2 is done by exponentiated the reward function, and normalizing it to sum to one. Note that this approach is actually solving a harder problem: it is solving the robust-reward control problem for a much larger set of reward functions that contains the original set of reward functions. Because this approach is solving a more challenging problem, we do not expect that it will achieve the optimal minimax reward. However, we emphasize that this approach may be easier to implement than fictitious play, which we compare against. Different from experiments in Sections 4.4 and 5.3, the "LowerBound + MaxEnt" approach assumes access to the full reward

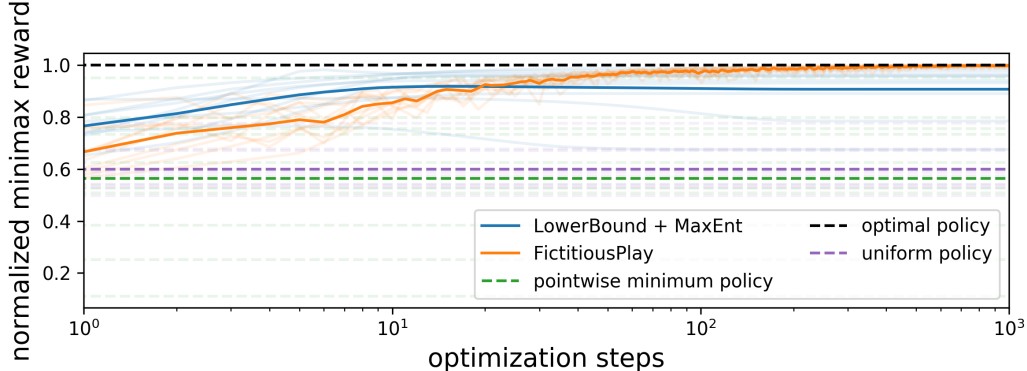

Figure 7: **Approximately solving an arbitrary robust-reward control problem.** In this experiment, we aim to solve the robust-reward control problem for an *arbitrary* set of reward functions. While we know that MaxEnt RL can be used to solve arbitrary robust-reward control problems exactly, doing so requires that we already know the optimal policy (§ B.5). Instead, we use the approach outlined in Section B.6, which allows us to *approximately* solve an arbitrary robust-reward control problem without knowing the solution apriori. This approach ("LowerBound + MaxEnt") achieves near-optimal minimax reward.

function, not just the rewards for the actions taken. For fair comparison, fictitious play will also use a policy player that has access to the reward function. Fictitious play is guaranteed to converge to the optimal minimax policy, so we assume that the minimax reward it converges to is optimal. We compare to two baselines. The "pointwise minimum policy" finds the optimal policy for a new reward function formed by taking the pointwise minimum of all reward functions: $\tilde{r}(a) = \min_{r \in R} r(a)$. This strategy is quite simple and intuitive. The other baseline is a "uniform policy" that chooses actions uniformly at random.

We ran each method on the same set of 10 robust-reward control bandit problems. In Figure 7, we plot the (normalized) minimax reward obtained by each method on each problem, as well as the average performance across all 10 problems. The "LowerBound + MaxEnt" approach converges to a normalized minimax reward of 0.91, close to the optimal value of 1. In contrast, the "pointwise minimum policy" and the "uniform policy" perform poorly, obtaining normalized minimax rewards of 0.56 and 0.60, respectively. In summary, while the method proposed for converting robust-reward control problems to MaxEnt RL problems does not converge to the optimal minimax policy, empirically it performs well.

### B.8    ALL ADVERSARIAL GAMES ARE MAXENT PROBLEMS

In this section, we generalize the previous result to show that MaxEnt RL can be used to solve to arbitrary control games, including robust control and regret minimization on POMDPs.

We can view the robust-reward control problem as a special case of a more general, two player, zero-sum game. The *policy player* chooses a policy at every round (or, more precisely, a mixture of policies). The second player, the *MDP player*, chooses the MDP with which we interact (or, more precisely, a mixture over MDPs). Formally, we define the set of policies $\Pi$ and MDPs $\mathcal{M}$, both of which correspond to *pure strategies* for our players. To represent *mixed strategies*, we use $\mathcal{Q}^\Pi \subseteq \mathcal{P}^\Pi$ and $\mathcal{Q}^\mathcal{M} \subseteq \mathcal{P}^\mathcal{M}$ to denote distributions over policies and models, respectively. Our goal is to find a Nash equilibrium for the following game:

$$\max_{p_\pi \in \mathcal{Q}^\Pi} \min_{p_\mathcal{M} \in \mathcal{Q}^\mathcal{M}} \mathcal{F}(p_\pi, p_\mathcal{M})$$

$$\text{where} \quad \mathcal{F}(p_\pi, p_\mathcal{M}) \triangleq \mathbb{E}_{\substack{\pi \sim p_\pi, \, \mathcal{M} \sim p_\mathcal{M} \\ a \sim \pi(a|s), \, s' \sim p_\mathcal{M}(s'|s,a)}} \left[ \sum_{t=1}^{T} r_\mathcal{M}(s_t, a_t) \right]. \tag{18}$$

The objective $\mathcal{F}$ says that, at the start of each episode, we *independently* sample a policy and a MDP. We evaluate the policy w.r.t. this MDP, where both the dynamics and reward function are governed

by the sampled MDP. We recall that the Nash Existence Theorem (Nash et al., 1950) proves that a Nash Equilibrium will always exist. With this intuition in place, we state our main result:

**Theorem B.5.** *For every solution adversarial control problem (Eq. 18), there exists a Markovian policy $\pi^*$ that is optimal:*

$$\exists \pi^* \ s.t. \min_{p_\mathcal{M} \in \mathcal{Q}^\mathcal{M}} \mathcal{F}(\mathbb{1}_{\pi^*}, p_\mathcal{M}) = \max_{p_\pi \in \mathcal{Q}^\Pi} \min_{p_\mathcal{M} \in \mathcal{Q}^\mathcal{M}} \mathcal{F}(p_\pi, p_\mathcal{M}).$$

In the statement of the theorem above, we have used $\mathbb{1}_{\pi^*}$ to indicate that the policy player uses a pure strategy, deterministically using policy $\pi^*$. While one might expect that mixtures of policies might be more robust, this theorem says this is not the case.

*Proof.* To begin, we note that $\mathcal{F}$ depends solely on the *marginal* distribution over states and actions visited by the policy. We will use $\rho_\pi(s, a)$ to denote this marginal distribution. Given the mixture over policies $p_\pi(\pi)$, each with its own marginal distribution, we can form the aggregate marginal distribution, $\bar{\rho}(s, a)$:

$$\bar{\rho}(s, a) = \int_\Pi p_\pi(\pi) \rho_\pi(s, a) d\pi.$$

We can now employ Ziebart (2010, Theorem 2.8) to argue that there exists a single policy $\pi^*$ with the same marginal distribution:

$$\exists \ \pi^* \ s.t. \ \rho_{\pi^*}(s, a) = \bar{\rho}(s, a) \ \forall s, a.$$

Thus, we conclude that

$$\min_{p_\mathcal{M} \in \mathcal{Q}^\mathcal{M}} \mathcal{F}(\mathbb{1}_{\pi^*}, p_\mathcal{M}) = \min_{p_\mathcal{M} \in \mathcal{Q}^\mathcal{M}} \mathcal{F}(p_\pi^*, p_\mathcal{M}).$$

$\square$

One curious aspect of this proof is that is does not require that we use MaxEnt policies. In Appendix B.9, we discuss how alternative forms of regularized control might be used to solve this same problem. We now examine four special types of adversarial games. For each problem, MaxEnt RL can be used to find the optimal Markovian policy.

> *Robust-Reward Control* – The robust-reward control problem (Definition 3.1) is a special case where the MDPs that the adversary can choose among, $\mathcal{M}$, have identical states, actions and dynamics, differing only in their reward functions. The adversary's choice of a distribution over MDPs is equivalent to choosing a distribution over reward functions.

> *POMDPs* – MaxEnt RL can be used to find the optimal Markovian policy for a POMDP. Recall that all POMDPs can be defined as distributions over MDPs. Let a POMDP be given, and let $p_\mathcal{M}^*$ be its corresponding distribution over MDPs. We now define the set of MDPs the adversary can among as $\mathcal{Q}^\mathcal{M} = \{p_\mathcal{M}^*\}$. That is, the adversary's only choice is $p_\mathcal{M}^*$. Note that the singleton set is closed under convex combinations (i.e., it contains all the required mixed strategies). Thus, we can invoke Theorem B.5 to claim that MaxEnt RL can solve this problem.

> *Robust Control* – Next, we consider the general robust control problem, where an adversary chooses both the dynamics and the reward function. To invoke Theorem B.5, we simply define a set of MDPs with the same state and actions spaces, but which differ in their transition probabilities and reward functions. Note that this result is much stronger than the robust-reward control problem discussed in Section B.5, as it includes robustness to dynamics.

> *Robust Adversarial Reinforcement Learning* – The robust adversarial RL problem (Pinto et al., 2017) is defined in terms of a MDP and a collection of "perturbation policies", among which the adversary chooses the worst. Note that the original MDP combined with perturbations from one of the perturbation policies defines a new MDP with the same states, actions, and rewards, but modified transition dynamics. Thus, we can convert the original MDP and collection of perturbation policies into a collection of MDPs, each of which differs only in its transition function.

While MaxEnt RL can find the optimal Markovian policy for each problem, restricting ourselves to Markovian policies may limit performance. For example, the optimal policy for a robust control problem might perform system ID internally, but this cannot be done by a Markovian policy.

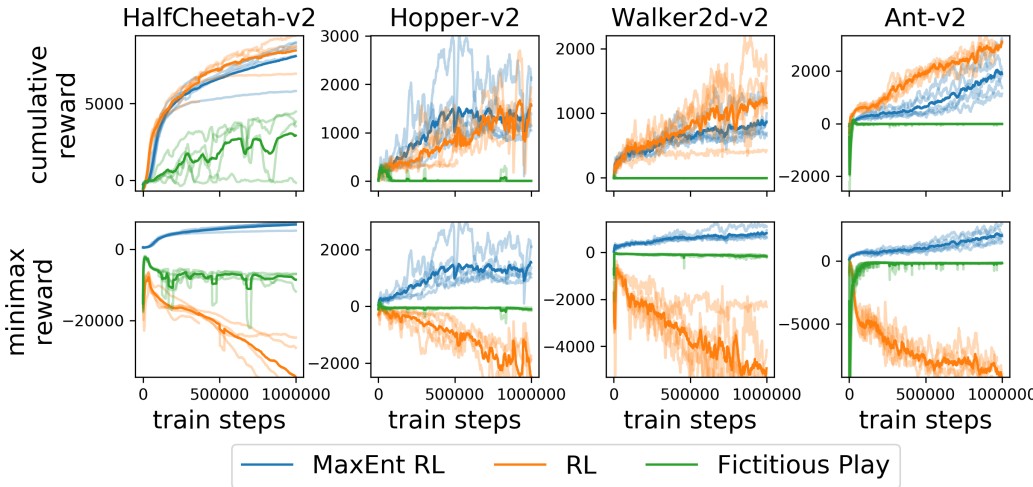

Figure 8: **MaxEnt RL solves robust-reward control problems for robotic control tasks.** In the top row, we plot the average cumulative reward, with each column showing a different environment. In the bottom row, we plot the average worst-case reward. While both MaxEnt RL and standard RL succeed at maximizing the cumulative reward, only MaxEnt RL succeeds at maximizing worst-case reward. The thick line is the average over five random seeds (thin lines).

### B.9 ALTERNATIVE FORMS OF REGULARIZED CONTROL

In this section, we examine why we used MaxEnt RL in Section B.5. If we write out the KKT stationarity conditions for the robust-reward control problem, we find that the solution to the robust-reward control problem $\pi$, is also the solution to a standard RL problem with a reward function that is a convex combination of the reward functions in the original set.

$$\nabla_\pi \mathbb{E}_\pi \left[ \sum_{t=1}^{T} \bar{r}(s,a) \right] \implies \pi \in \arg\max_\pi \mathbb{E}_\pi \left[ \sum_{t=1}^{T} \bar{r}(s_t, a_t) \right],$$

where

$$\bar{r}(s,a) \triangleq \sum_i \lambda_i r_i(s,a).$$

Above, we have used $\lambda_i$ as the dual parameters for reward function $r_i$. However, we have no guarantee that $\pi$ is the *unique* solution to the RL problem with reward function $\bar{r}$. Using MaxEnt RL guarantees that the optimal policy is unique.[2] More broadly, we needed a regularized control problem. It is interesting to consider what other sorts of regularizers can induce unique solutions, thereby allowing us to reduce robust-reward control to these other problems as well. We leave this to future work.

## C ADDITIONAL EXPERIMENTS

In Section 5, we argued that MaxEnt RL was equivalent to solving a robust-reward control problem. The proof of Theorem 5.2 holds for arbitrary MDPs, not just bandits. In this section, we evaluate MaxEnt RL and two baselines on four robotic control tasks, and find that MaxEnt RL optimizes worst-case reward better than the baselines.

We consider continuous control tasks taken from the standard OpenAI Gym (Brockman et al., 2016) benchmark: `HalfCheetah-v2`, `Hopper-v2`, `Walker2d-v2`, and `Ant-v2`. Each task has a maximum horizon of 1,000 steps. We used SAC (Haarnoja et al., 2018b) as our MaxEnt RL algorithm and SVG (Heess et al., 2015) as our standard RL algorithm. We also compared with a version fictitious play that modified SVG to choose the worst-case reward function at each time step. For computational reasons, we do not include historical averaging for the fictitious play baseline in this experiment. Following Haarnoja et al. (2018b), we use a temperature of 1/5 (see Appendix B.4).

---

[2]However, the parameters of the optimal policy may not be unique if there is a surjective mapping from policy parameters to policies.

We used the same hyperparameters for each algorithm, taking the hyperparameters directly from the TF-Agents (Guadarrama et al., 2018) implementation of SAC. To evaluate each algorithm, we computed both the average cumulative reward, as well as the average worst-case reward. The worst-case reward was defined as in Equation 2. As shown in Figure 8, both MaxEnt RL and standard RL can maximize the cumulative reward, but only MaxEnt RL succeeds as maximizing the worst-case reward. In summary, this experiment supports our proof that MaxEnt RL is solving a robust-reward control problem in arbitrary MDPs.

## D  AN EXAMPLE META-POMDP

Consider a simple MDP shown to the right. There are two actions (Up and Down), five states (1, 2, 3, 4, 5) and episodes last two steps (i.e., two actions). We will assume that there is only one action that can be taken in the final state, and will therefore ignore the last action. The agent always starts in state 1. Now, we define a meta-POMDP for this MDP. We define the trajectory distribution as

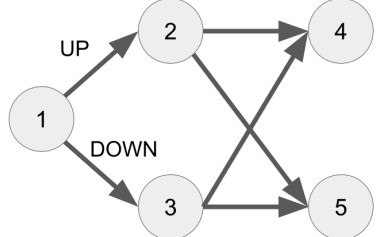

$$p(\tau^*) = \frac{1}{2}\delta(\tau^* = (1 \to 2 \to 4)) + \frac{1}{2}\delta(\tau^* = (1 \to 2 \to 5))$$

**Meta-episodes and meta-steps**: Below, we go through an example meta-episode.

> **Meta-Episode 1**: At the start of a meta-episode, a trajectory $\tau^* \sim p(\tau^*)$ is sampled. We will say that we sampled $\tau^* = (1 \to 2 \to 4)$.
>
> > **Meta-step 1**: One meta-step corresponds to one episode in the MDP (i.e., two actions). For example, a meta-step might consider of the policy choosing actions Up and Down, resulting in trajectory $\tau = (1 \to 2 \to 5)$. Since the trajectory did not equal the target trajectory ($\tau \neq \tau^*$), the agent receives a reward of -1 (for the meta-step) and the meta-episode does not terminate. Note that the reward and termination are based on the (unknown) target trajectory $\tau^*$, not on the trajectory distribution $p(\tau^*)$.
> >
> > **Meta-step 2**: In the second meta-step, we will say the policy chooses actions Up and Up, resulting in trajectory $\tau = (1 \to 2 \to 4)$. Since the trajectory does equal the target trajectory ($\tau = \tau^*$), the agent receives a reward of 0 (for the meta-step) and the meta-episode terminates. The cumulative return for the agent in this meta-episode is: -1 (for meta-step 1) + 0 (for meta-step 2) = -1. The optimal policy would have immediately chosen $\tau^*$ and received a reward of 0, so our policy has a regret of $0 - (-1) = 1$.
>
> **Meta-episode 2**: At the start of the next meta-episode, another trajectory $\tau^* \sim p(\tau^*)$ is sampled. We continue as in meta-episode 1.

**Regret**: For this task, the optimal policy $\pi_1$ chooses uniformly between trajectory $(1 \to 2 \to 4)$ and trajectory $(1 \to 2 \to 5)$. The regret of this policy is

$$\text{Regret}_p(\pi_1) = p(\tau^* = (1 \to 2 \to 4))\frac{1}{\pi_1(\tau^* = (1 \to 2 \to 4))} + p(\tau^* = (1 \to 2 \to 5))\frac{1}{\pi_1(\tau^* = (1 \to 2 \to 5))}$$

$$= \frac{1}{2}\frac{1}{1/2} + \frac{1}{2}\frac{1}{1/2} = 2.$$

As another example, consider a policy that chooses uniformly between three trajectories: $(1 \to 2 \to 4)$, $(1 \to 2 \to 5)$, and $(1 \to 3 \to 4)$. The regret of this (suboptimal) policy $\pi_2$ is:

$$\text{Regret}_p(\pi_2) = p(\tau^* = (1 \to 2 \to 4))\frac{1}{\pi_2(\tau^* = (1 \to 2 \to 4))} + p(\tau^* = (1 \to 2 \to 5))\frac{1}{\pi_2(\tau^* = (1 \to 2 \to 5))}$$

$$= \frac{1}{2}\frac{1}{1/3} + \frac{1}{2}\frac{1}{1/3} = 3.$$

**Solving the meta-POMDP**: In Section 4.2, we said that the optimal policy should satisfy $\pi(\tau) \propto \sqrt{p(\tau^*)}$. In this example, this identity means that the optimal policy should be $\pi(\tau) = \frac{1}{2}\delta(\tau^* =$

$(1 \to 2 \to 4)) + \frac{1}{2}\delta(\tau^* = (1 \to 2 \to 5))$. Note that the optimal policy, $\pi_1$ satisfies this identity, while the suboptimal policy, $\pi_2$, does not.

**Lemma 4.1**: Lemma 4.1 says that there should exist a reward function such that that MaxEnt RL problem with this reward function and meta-POMDP have the same solution. In this simple example, one such reward function is

$$r(s, a) = \begin{cases} -\infty & \text{if } s = 3 \\ 0 & \text{otherwise} \end{cases}.$$

Since the dynamics in this simple example are deterministic, the optimal MaxEnt RL policy chooses trajectory $\tau$ with probability

$$pi^*(\tau) \propto e^{\sum_t r(s_t, a_t)} = \begin{cases} 1 & \text{if } \tau \in \{(1 \to 2 \to 4), (1 \to 2 \to 5)\} \\ 0 & \text{otherwise} \end{cases}.$$

Thus, MaxEnt RL on this reward function yields the optimal policy for the meta-POMDP.

**Goal-reaching meta-POMDPs**: Finally, we consider a goal-reaching task. For example, we will say that the task is to reach goal $s_T = 4$. We define the trajectory distribution as uniform over all trajectories that reach this goal at the final time step:

$$p(\tau^*) = \frac{1}{2}\delta(\tau^* = (1 \to 2 \to 4)) + \frac{1}{2}\delta(\tau^* = (1 \to 3 \to 4)).$$

This trajectory distribution satisfies the assumption of Lemma 4.2, which states that the density can be written as a function of the last state in the trajectory:

$$p(\tau^*) = \tilde{p}(s_T(\tau), a_T(\tau)) = \frac{1}{2}\delta(s_T(\tau) = 4).$$

Following Lemma 4.2, the corresponding MaxEnt RL reward function is

$$r(s_t, a_t) = \frac{1}{2}\delta(t = T)\log\tilde{p}(s_T(\tau), a_T(\tau)) = \begin{cases} -\infty & \text{if } t = T \text{ and } s_T(\tau) \neq 4 \\ 0 & \text{otherwise} \end{cases}.$$

Since the dynamics in this simple example are deterministic, the optimal MaxEnt RL policy chooses trajectory $\tau$ with probability

$$\pi^*(\tau) \propto e^{\sum_t r(s_t, a_t)} = \begin{cases} 1 & \text{if } \tau \in \{(1 \to 2 \to 4), (1 \to 3 \to 4)\} \\ 0 & \text{otherwise} \end{cases}.$$

The "magic" of Lemma 4.2 is that we can obtain the optimal policy for this meta-POMDP by applying MaxEnt RL to a reward function that does provide reward whenever the agent reaches goal state $s_T$, regardless of the trajectory taken.

