# OpenReview forum: "If MaxEnt RL is the Answer, What is the Question?"
_ICLR.cc/2020/Conference — Reject_

### Official Review · AnonReviewer2 · 2019-10-16
**Official Blind Review #2**

**Rating:** 8

**Review:**

The paper investigates the reason behind the success of MaxEntropy in reinforcement learning theoretically, connecting it to  robust control.

I  think the paper should be accepted, as it investigates an important approach and offers useful insight. I think that the robust reward is an interesting perspective and the paper is also well written.

I need to remark that I am not familiar enough with RL theory literature to know of novel this work is,.

Detailed remarks:
- There is an error in the proof in A.1. You are optimizing a functional not a function, so the cannot simply use Lagrange multipliers (also the derivation ignores the integral). The problem is solved easily with (constrained) Euler-Lagrange and must be corrected.
- I disagree with the exploration paragraph in sec. 2. While the final applied agent might be deterministic, using a stochastic agent for exploration during learning is helpful. Exploration might not be only (or main) motivation behind MaxEntRL but it still a good motivation.
-  I was happy to see the limitation of lemma 4.1 stated clearly, as some paper are less honest with their limitations.
- You write "Only the oracle version of fictitious play, which makes assumptions not made by MaxEnt", maybe I missed it but I didn't see what assumptions the oracle made.
- MaxEnt has been very successful in inverse RL where we try to find the reward, which seems connected to the conclusions here about robustness to reward perturbations. While adding analysis on IRL might be outside the scope of this paper, something at least should be said about maxEnt and IRL and the connection to the current results.

Typo:
"inference problem be defining" -> "inference problem by defining"

**Experience Assessment:**

I have read many papers in this area.

**Review Assessment: Checking Correctness Of Derivations And Theory:**

I carefully checked the derivations and theory.

**Review Assessment: Checking Correctness Of Experiments:**

I assessed the sensibility of the experiments.

**Review Assessment: Thoroughness In Paper Reading:**

I read the paper thoroughly.

---

> ### Author Response · Authors · 2019-11-09
> **Author Response**
>
> Thanks for the review and feedback for improving the paper!
> 1. Typo in proof in A.1: Thanks for pointing this out this typo! We should be optimizing w.r.t. \pi(\tau), so the corresponding derivative should be dL/d\pi(\tau). We have fixed this typo and clarified that we are using constrained Euler-Langrange.
> 2. Exploration: Yes, we agree that MaxEnt RL empirically performs good exploration. We have added a sentence to the Exploration paragraph in Section 2 to not dismiss this motivation.
> 3. Oracle fictitious play: The fictitious play baseline has access to the rewards for all arms, not just the arm it selected (we clarified this halfway through the prior paragraph.). In contrast, the MaxEnt RL method only observes the rewards for the arm that it actually pulls.
> 4. MaxEnt IRL: Thanks for this great suggestion! IRL is an ill-posed problem, so it lends itself quite naturally to approaches that can cope with reward variability. We conjecture (but have not proven) that MaxEnt IRL yields reward functions which, when combined with MaxEnt RL, guarantee that the resulting policy perform well on the unknown reward function. We have added some discussion of this to the conclusion.

---

### Official Review · AnonReviewer1 · 2019-10-23
**Official Blind Review #1**

**Rating:** 3

**Review:**

Summary :
The paper discusses the use of maximum entropy in Reinforcement Learning. Specifically, it relates the solution of the maximum entropy RL problem to the solutions of two different settings, 1) a ‘meta-POMDP’ regret minimization problem and 2) a ‘robust reward control’ problem. Both cases follow with simple experiments.

I feel the paper could have been written more clearly. There seem to be too many definitions and descriptive examples that diverge the attention of the reader from the main problem setting. There are quite a bit of grammatical errors in the paper, making it even harder to follow. With these many definitions in the text, it is hard to make out the actual contributions of the work. Moreover, the experiments are restricted to the bandit setting and do not provide any empirical evidence on the MDP centered theory. Overall, although the paper does well in motivating the problem, the lack of rigorous experiments and poorly structured writing advocate for a weak rejection.


Comments/questions:
- Can the authors comment on why it makes intuitive sense to study the meta-POMDP and robust reward control problem settings together? I see the commonality being the reward variability, but is there something else?
- If one wants to solve the meta-POMDP through max entropy RL, how general/strong is the assumption that we are given access to the target trajectory belief?
- In the goal reaching meta-POMDP, it makes sense to only have the final state distribution in the definition. What does the action taken in the final state signify?
- It would be more intuitive to note the optimal solution as pi* and not pi (Lemma 4.1).
- In the meta-POMDP, does the task change after every meta-episode?
- I think it would be better to have separate, consistently named subsections devoted to defining the two problem settings and then move on to proving equivalence with the max entropy case.


**Experience Assessment:**

I have published one or two papers in this area.

**Review Assessment: Checking Correctness Of Derivations And Theory:**

I assessed the sensibility of the derivations and theory.

**Review Assessment: Checking Correctness Of Experiments:**

I assessed the sensibility of the experiments.

**Review Assessment: Thoroughness In Paper Reading:**

I read the paper thoroughly.

---

> ### Author Response · Authors · 2019-11-09
> **Author Response**
>
> Thank you for the response. We appreciate the feedback and will work to incorporate it into the paper. As suggested, we ran additional experiments on four benchmark robotic control tasks from OpenAI Gym: HalfCheetah-v2, Walker2d-v2, Hopper-v2, and Ant-v2. These tasks have horizons up to 1000 steps, so they are useful for evaluating our claims about MaxEnt RL on long-horizon tasks. In our experiments, we find that while both standard RL and MaxEnt RL can maximize reward, only MaxEnt RL effectively maximizes worst-case reward. Fictitious Play fails to maximize worst-case reward on these tasks. We have added a new section to the Appendix (Appendix C) to include figures and a summary from this experiment. We believe that these experiments address the concern that the experiments were solely on bandit settings.
>
> We have clarified the paper in the following ways. We have clarified the notation for \pi* in the proof of Lemma 4.1 and added subsection headers to Section 3 to clarify that we introduce the two problem settings there. In Section 3.3, we highlight that these two problem settings are similar in that they both involve variability in the reward function. While these are the two settings that we know are equivalent to MaxEnt RL, we speculate that there are other equivalent problems. We leave this to future work. We have also clarified the paper by incorporating the suggestions from Reviewer #3. We believe that this addresses the concerns about clarity, but would welcome additional suggestions.
>
> Clarifications
> 1. Trajectory belief: There may be a misunderstanding here — we do not make any additional assumptions, as the trajectory belief is derived direction from the reward function (and vice-versa).
> 2. Action in the last state: For notational simplicity, we included an action for every state so that we can say that the reward function is evaluated at each state-action pair. As there are no future states, the only effect of this action is in determining the reward given at the final step. If the reward function is defined solely in terms of the state, then we could ignore the action at the last state. Including the action at the last state means that we can view bandits as a special case, where there is only one state, and rewards are defined solely in terms of actions.
> 3. Task sampling in the meta-POMDP: Yes, a new task is sampled i.i.d. at the start of each meta-episode. Of the distribution over tasks is finite, then it is possible that the same task is sampled for two consecutive meta-episodes.

---

### Official Review · AnonReviewer3 · 2019-10-24
**Official Blind Review #3**

**Rating:** 3

**Review:**

This paper aims to theoretically understand the reason that MaxEnt RL (RL with an entropy bonus) works so well. It suggests that MaxEnt RL works well in the setting where there is uncertainty about the reward function. It proves two main theorems to support this. First, for any instance of a class of “meta-POMDPs” where the agent only has a *belief* over the goal trajectory, there exists a reward function for which MaxEnt RL on that reward function is the optimal solution to the meta-POMDP. Second, for any reward and MDP, there exists a set of reward functions such that MaxEnt RL maximizes the worst-case return for a reward chosen from that set.

The paper tackles an important question, since entropy bonuses are commonly used in RL, but are primarily a “hack” added to incentivize exploration without any principled justification. However, I question the applicability of the theorems to the success of MaxEnt RL.

I’m recommending a weak reject, for the following reasons, in order of importance (explained in more detail later):
- Section 4.2 assumes that you must deploy a single policy across all timesteps of the meta-POMDP, but it should be possible to update your policy across timesteps.
- Section 4.3 claims to reduce goal-reaching problems to single-goal-trajectory problems, but this doesn’t work because goal-reaching problems can have multiple optimal trajectories.
- The adversary of Theorem 5.2 is quite unusual, and it is unclear why robustness to such an adversary should be useful.
- Since the theorems are about cases where the agent is uncertain about the reward, they don’t explain why MaxEnt RL is useful even in the case where we have no adversaries and care only about performance on a single reward function.
- The paper is not very clear. The quality of exposition could be improved significantly.

It seems quite likely that some of my critiques are misguided (especially the one about Section 4.2), and I encourage the authors to point this out in the rebuttal.

----

My primary complaint is that these theorems don’t apply to the case they are meant to explain: the authors say they want to explain why MaxEnt RL works in practice, but their explanations and theorems center on cases in which the reward is unknown. But currently MaxEnt RL works well on tasks where the reward is known! It’s not that MaxEnt RL finds different solutions that don’t do well on the original reward but do better by some other criterion: the solutions found by MaxEnt RL are the best solutions when evaluated by the known reward.

Nonetheless, it is still an interesting question to study the benefits of MaxEnt RL in the context of reward uncertainty. I’d recommend that the authors change the motivation and introduction to focus on that setting, without claiming to explain why MaxEnt RL works with current systems. For the rest of the review, I’ll evaluate the paper from that perspective.

----

I found the section on meta-POMDPs very confusing. First, one minor confusion: I believe you are assuming that the underlying MDP is deterministic? (Perhaps not, but the assumption that there is a policy with probability proportional to sqrt(p(tau)) is much less likely to hold in stochastic environments.)

Section 4.2 is confusing to me. First, let me quote the description of the meta-POMDP: “Each meta-step of the meta-POMDP corresponds to one episode of the original MDP. A meta-episode is a sequence of meta-steps, which ends when the agent solves the task in the original MDP. Intuitively, each meta-episode in the meta-POMDP corresponds to multiple trials in the original MDP, where the task remains the same across trials. The agent keeps interacting with the MDP until it solves the task.”

By my understanding, this means that in the meta-POMDP the agent is able to learn across episodes. In particular, in the setting where the agent has a distribution over a single goal trajectory, and the underlying MDP is deterministic, the optimal policy is obvious: try the most probable goal trajectory, then try the second most probable goal trajectory, and so on until you find the true goal trajectory, and then repeatedly execute that goal trajectory. Why is this not the result of Section 4.2?

(My guess is that you assumed that the agent is not able to learn across episodes (why?), or that the agent receives literally zero information about whether or not it has completed the goal (highly unrealistic, and certainly doesn’t apply to the physician example). Perhaps there is a different unstated assumption instead.)

I’d encourage the authors to more clearly formalize the meta-POMDP as well as the relevant assumptions that lead to the results of Section 4.2. As currently defined, I don’t agree with Section 4.1, though I do think there is a formalization which makes everything work (though that formalization does not seem realistic to me).

----

I’m also confused about Section 4.3. In Section 4 (before subsections) and Section 4.1, it seems that the model is that there is a *single* goal trajectory which the agent must replicate, but the agent is uncertain about that trajectory. However, Section 4.3 considers goal-reaching problems, in which there can be *multiple* goal trajectories. I don’t understand how the goal-reaching problem is reduced to the single-goal-trajectory problem: I believe the current reduction is *not* solving a goal-reaching problem.

For simplicity, let’s consider a goal-reaching problem where the goal is known: a 3x3 grid where the agent starts at the bottom left and the goal is to get to the top right. Assume a horizon of 4 for simplicity. Then any trajectory involving two Ups and two Rights solves the problem, i.e. there are six optimal trajectories: {UURR, URUR, URRU, RUUR, RURU, RRUU}. However, the meta-POMDP defined in the reduction assigns 1/6 probability to each such trajectory, which by the semantics of the meta-POMDP means that the agent “actually” wants to choose one of those trajectories in particular, but doesn’t know which one is appropriate. This is not an accurate representation of the goal-reaching problem: the goal-reaching problem is solved as long as *any* of these trajectories are selected.

(Lemma 4.2 is technically accurate, because it only talks about the meta-POMDP and the corresponding reward function, and doesn’t claim anything about the relation to the goal-reaching problem, but all the prose around Lemma 4.2 is misleading.)

This general problem persists even if you have a belief over the goal state, rather than being certain about the goal state. You could try fixing this by changing the meta-POMDP to put probability over *sets* of goal trajectories, and Section 4.1 would still go through (the regret would be a geometric random variable of the probability the trajectory lies within the true goal trajectory set), but Section 4.2 would no longer work.

----

Theorem 5.2 shows that there exists a set of reward functions such that MaxEnt RL maximizes the worst-case return for a reward chosen from that set. Taken literally, this is not particularly interesting, as it is also true of regular RL: the optimal regular RL policy pi optimizes the worst-case return for the set { r’(s, a) = r(s, a) + f(s, a) with f in F }, for any class of non-negative functions F that contains the zero function, simply because every other reward in the set is at least as large as the true reward r(s, a). (Another set that works is { r’(s, a) = K r(s, a) with K > 0 }.)

So really the interesting content of the second theorem is the particular set of reward functions that MaxEnt RL is robust to: the set {r’(s, a) = r(s, a) - log q(a | s) for all q in Pi}. This is an interesting adversary model: essentially, at every state, the adversary is required to provide a non-negative bonus b_a for each action a. To prevent the adversary from providing zero bonus everywhere, the bonuses must satisfy \sum_a exp(b_a) = 1.

This essentially means that at each state, the adversary wants to find actions that the policy doesn’t take, and allocate more of the bonus to that action. Given this particular adversary, it makes sense to inject a little noise into the policy so that the adversary can’t “hide” the bonus in actions that are rarely taken, and so it makes sense the MaxEnt RL could be the best response to such an adversary. However, this is quite a strange adversary -- we wouldn’t usually expect that the things we need to be robust to are going to be dividing up some bonus across actions. So I’m not very confident that this theorem will actually matter in practice.

----

Typos:

Abstract: “Probability, as this strategy is called,”: “Probability matching”
Introduction: “model for decision decision making”: repeated “decision”
Introduction: “empirical benefits of MaxEnt RL arise implicitly solving”: “arise by implicitly solving”
Preliminaries: “These approaches cast optimal control as an inference problem be defining”: “be” --> “by”
Preliminaries: In the derivation of Eq. 1, p(s_1) was dropped
Preliminaries: The last equality in the derivation of Eq. 1 would only hold if you had a log on the LHS, so you probably wanted to maximize log p(O_t) in the entirety of the derivation.
Lemma 4.2: “p˜(sT(τ), aT(τ)) = p(τ)”: Reverse the order to write “p(τ) = p˜(sT(τ), aT(τ))” to follow the convention that the thing being defined is on the LHS

**Experience Assessment:**

I have published one or two papers in this area.

**Review Assessment: Checking Correctness Of Derivations And Theory:**

I assessed the sensibility of the derivations and theory.

**Review Assessment: Checking Correctness Of Experiments:**

I did not assess the experiments.

**Review Assessment: Thoroughness In Paper Reading:**

I read the paper thoroughly.

---

> ### Author Response · Authors · 2019-11-09
> **Author Response**
>
> Thank you for the incredibly detailed review and the suggestions for improvement. We hope to persuade you that the setting we consider is actually quite reasonable. The assumption that the policy cannot improve within a single meta-episode is just a Markovian assumption; the policy can (and should) be improved between meta-episodes. You are right to note that our assumptions are conservative (i.e., pessimistic), so our analysis serves to lower bound performance in settings with more structure (e.g., providing rewards at every time step) with learning algorithms that could be more powerful (e.g., equipped with memory). We don't assume deterministic dynamics. While the meta-POMDP is well defined for arbitrary target distributions, our proof that MaxEnt RL solves the meta-POMDP does assume that there exists a policy that matches the target distribution. Note that this distribution can be defined in terms of feasible behaviors (e.g., as a mixture of previously observed behaviors), guaranteeing that it is possible to match this distribution.
>
> We believe that our analysis is useful for understanding precisely what optimization problem MaxEnt RL is solving. While it remains an open question why MaxEnt RL works well in the single-task setting, we speculate that this paper is a step towards answering that question. We hypothesize MaxEnt RL may work because optimizing the worst-case reward might lead to good exploration (because the agent would seek to avoid previously observed, low-reward states) and might stabilize off-policy learning (because the actor would not purely exploit actions to which the critic spuriously assigned high value).
>
> We believe that the reduction from a goal-reaching meta-POMDP to a trajectory-reaching meta-POMDP is valid. If the task is to reach a particular goal state, we define the trajectory distribution as placing equal probability on all trajectories that end in this goal. A policy that matches the target trajectory distribution by choosing uniformly among these trajectories will always arrive at the goal state. You are correct that there may be many trajectories that reach a goal, and this observation highlights a key difference between standard RL and MaxEnt RL: any policy that reaches the goal is optimal for the standard RL objective, but only the policy that is uniform over all goal-reaching trajectories is optimal for the MaxEnt RL objective. MaxEnt RL can therefore been seen as resolving ambiguity about which policy to choose. Our analysis suggests that the single policy chosen by MaxEnt RL is not arbitrary, but rather is one with desirable robustness properties.
>
> We have clarified the paper with the following updates. We have fixed the typos mentioned and reworded paragraph 3 in the Introduction to emphasize that our analysis does not explain why MaxEnt RL works in the single task setting. We have added a paragraph to Section 4.1 to clarify the assumption that the policy does not change between meta-steps. We added an explanation of why the adversary in Theorem 5.2 is reasonable in the paragraph following this theorem. We believe that these modifications address all of the writing concerns, though we would welcome additional suggestions on how to clarify the paper.

---

> > ### Comment · AnonReviewer3 · 2019-11-11
> > **What are the semantics of p(τ)?**
> >
> > What are the semantics of the belief over goal trajectories $p(\tau)$? Based on the sentence
> >
> > > We will use the most general definition of success as simply matching some target trajectory, $\tau^*$.
> >
> > I assumed that there was a _single_ $\tau^*$ that gets reward, and every other τ gets zero reward, and the agent is uncertain about which one is the one that gets reward. But then I don't understand this part of your reply (emphasis mine):
> >
> > > We believe that the reduction from a goal-reaching meta-POMDP to a trajectory-reaching meta-POMDP is valid. [...] You are correct that there may be _many_ trajectories that reach a goal
> >
> > This seems to directly contradict the assumption that there's a _single_ $\tau^*$ that gets reward.
> >
> > Alternatively, maybe the semantics are that _every_ $\tau$ that $p(\tau)$ puts support on gets reward, and the goal is simply to have some kind of maximum entropy. This still isn’t completely defined (what’s the difference between $p(\tau) = 0.25$ and $p(\tau) = 0.5$?) but there’s already a problem — under these semantics, the regret would not be $E[\frac{1}{\pi(\tau^*)}]$, the regret would be zero, since the policy would always choose one of the trajectories in $p(\tau)$ and so would always get reward.
> >
> > It might be helpful for my understanding if you construct an explicit toy example of a goal-state-based meta-POMDP, and said what the meta-steps, meta-episodes, $p(\tau)$ etc. were, and showed how the theorems you prove in Sections 4.1, 4.2 and 4.3 apply to this meta-POMDP.

---

> > > ### Author Response · Authors · 2019-11-12
> > > **Clarifying semantics of p(τ) and a new toy example (Appendix D)**
> > >
> > > (I think) Your understanding is correct: in the meta-POMDP, there is a single trajectory $\tau^*$ that gets reward. For goal-reaching tasks, there might exist multiple trajectories $\tau$ that reach a particular goal state $s_T$. The goal-reaching meta-POMDP will sample one of these trajectories $\tau^*$ that ends at the goal state $s_T$. The agent will receive reward if it exactly matches $\tau^*$, but will not receive reward for any other trajectory, including one that reached the goal state $s_T$ via another path. The optimal policy is one that chooses uniformly among trajectories that end at goal state $s_T$. Note that, under this notion of optimality, a policy that always reaches the goal but uses a fixed trajectory to get there is not optimal.
> > >
> > > To clarify the notation and results for the meta-POMDP, we have added a new Appendix D. Here, we  introduce a simple meta-POMDP, describe what meta-episodes, meta-steps, and regret mean in this simple meta-POMDP, and then discuss how the results from Section 4 apply to this meta-POMDP. We hope this clarifies the meta-POMDP definitions and results. Please let us know if anything is unclear.

---

> > > > ### Comment · AnonReviewer3 · 2019-11-12
> > > > **Thanks**
> > > >
> > > > Thanks, Appendix D, combined with the Markov assumption (that I still find very unintuitive) was helpful in understanding the point of the goal-reaching section.

---

> > ### Comment · AnonReviewer3 · 2019-11-11
> > **Why is the Markov assumption reasonable?**
> >
> > Given the Markov assumption for the meta-POMDP, I believe the results in Sections 4.1 and 4.2, but it’s a really odd assumption. Let’s consider the physician example:
> >
> > > Each meta-step corresponds to one visit to the physician, which might entail running some tests, performing an operation, and prescribing a new medication. The meta-episode is the sequence of patient visits, which ends when the patient is cured.
> >
> > And now let’s add in the Markov assumption:
> >
> > > Our analysis of the meta-POMDP will consider policies that are Markovian within a meta-episode: while the policy can be updated between meta-episodes, the policy cannot use information from one meta-step to take better actions in a future meta-step within the same meta-episode
> >
> > So this means that every time the patient returns to the physician, the physician “forgets” everything (s)he has learned about the patient, and tries something essentially at random (or rather, based on priors). This is obviously going to be a terrible policy. I don’t see why we should infer anything from the fact that MaxEnt RL leads to this policy for some reward function.
> >
> > (More broadly, with any real physician, I generally hope that their optimal policy is not randomized -- why on earth should it be? Even if they don't know what would cure me, they should at least have a best guess, and / or some best information to collect, and they should deterministically choose that.)
> >
> > > Our results will therefore be lower bounds on the performance of non-Markovian policies.
> >
> > Assuming that the semantics of $p(\tau)$ are that there is a _single_ trajectory that gets reward (see previous comment for more details), we know what the optimal non-Markovian policy is — as I said in my original review:
> >
> > > try the most probable goal trajectory, then try the second most probable goal trajectory, and so on until you find the true goal trajectory, and then repeatedly execute that goal trajectory.
> >
> > Suppose that $p(\tau)$ has support over $n$ trajectories. My guess is that this non-Markovian policy gets $O(n)$ regret, while the optimal Markovian policy of Section 4.2 gets $O(n \log n)$ regret, though I haven’t proven this. (My guess is that these bounds become tight when $p(\tau)$ is uniform over the $n$ trajectories.)

---

> > > ### Author Response · Authors · 2019-11-12
> > > **MaxEnt RL might not be reasonable, but understand it is important.**
> > >
> > > You're absolutely correct that memoryless policies are not optimal for solving the meta-POMDP (We likewise hope our physicians are not memoryless.). I think your analysis is right. Enumerating over $n$ items will get us to the correct trajectory in $O(n)$ meta-steps while sampling uniformly at random will get us the correct trajectory in $O(n \log(n))$ meta-steps (It's an instance of the coupon collector problem.).
> > >
> > > So, why is this a reasonable setting to consider? Given that MaxEnt RL is an active area of research today, we believe that it is important to explain the problems settings for which MaxEnt RL is optimal. The somewhat unusual assumptions in these problems suggest that MaxEnt RL might be an unreasonable algorithm, and there may exist control algorithms that solve problems with fewer unusual assumptions. Nonetheless, we believe that explaining a widely used algorithm is a useful contribution.

---

> > > > ### Comment · AnonReviewer3 · 2019-11-12
> > > > **It doesn’t seem suggestive that MaxEnt RL is unreasonable either**
> > > >
> > > > Originally, I thought you were claiming:
> > > >
> > > > Claim: MaxEnt RL is a good algorithm because it gains robustness by optimizing worst-case reward under reward uncertainty.
> > > > or
> > > > Claim: MaxEnt RL is a good algorithm in settings where you are uncertain about the reward, because it is optimal for these settings with uncertain rewards.
> > > >
> > > > To be convinced of either of these claims, I would want to be convinced that the settings under which MaxEnt RL optimizes worst-case reward are reasonable settings. I'm not convinced of this with the meta-POMDP explanation since the Markov assumption does not seem reasonable to me. I'm unsure with the result of Section 5 -- being optimal under an adversary is compelling, but it may only hold for a very specific kind of adversary that doesn't generalize. In both cases, I'm unsure because the space of reward functions very large: saying that an algorithm optimizes some particular subset of reward functions may not be saying very much, just because there are _so many_ reward functions that a lot (though not all) behavior can be rationalized as optimizing some reward function or subset of reward functions.
> > > >
> > > > Now it seems you are arguing for a different claim:
> > > >
> > > > Claim: MaxEnt RL is an unreasonable algorithm.
> > > >
> > > > I certainly agree that if true, this would be an important claim that the community should know about.
> > > >
> > > > To be convinced of this claim, I would want to be convinced that there is _no_ reasonable problem setting which MaxEnt RL is optimal for, _and_ that there is no other explanation (such as exploration) which MaxEnt RL is good for.
> > > >
> > > > This is a very hard claim to argue for -- proving a negative is very difficult. In particular, showing that MaxEnt RL is optimal in unreasonable settings is hardly any evidence for this claim -- why couldn't it be optimal in reasonable settings as well, that we haven't yet figured out?
> > > >
> > > > > there may exist control algorithms that solve problems with fewer unusual assumptions
> > > >
> > > > A demonstration of such an algorithm would be very compelling, but for the reasons above I don't think the evidence in this paper strongly suggests that such algorithms exist. (Nor do they suggest that such algorithms don't exist -- they just aren't much evidence either way.)
> > > >
> > > > > we believe that explaining a widely used algorithm is a useful contribution.
> > > >
> > > > I certainly agree, but "explaining" should involve a claim of the form "MaxEnt RL could never work because ..." or "MaxEnt RL works in situations X because ..." (where situations X are similar to the ones where MaxEnt RL works in practice). I've explained above why I don't think either of these claims are supported.
> > > >
> > > > For example, suppose I show that given census data, CNNs work well under the assumption that age and socioeconomic status are grouped together*. Is this evidence that CNNs are reasonable, or evidence that CNNs are unreasonable (since they require very particular groupings)? I think it's basically not evidence either way: the age-socieconomic-status-grouping assumption is a really strange and unrealistic assumption, so it doesn't explain why CNNs work in other settings like image recognition, but it also doesn't mean that CNNs are unreasonable, since there may be other reasonable settings or assumptions in which CNNs work well (as is actually the case).
> > > >
> > > > The meta-POMDP explanation in this paper seem qualitatively similar to me: the Markov assumption seems unreasonable, in the same way that age-socioeconomic-status-grouping seems unreasonable in the CNN case.
> > > >
> > > > Footnotes:
> > > >
> > > > *This might happen because there's a nonlinear effect between age and socioeconomic status that the CNN can only capture if the two are close together so that a single convolution operates on both of them together, but in this hypothetical I haven't figured that out.
> > > >
> > > > Side note:
> > > >
> > > > It isn't obviously isomorphic to the coupon collector problem, since in the coupon collector problem you need to collect _all_ of the coupons (trajectories), whereas here you need to consider the expected number of meta-steps needed to collect a _randomly selected_ coupon.
> > > >
> > > > Actually, now that I write that down, when $p(\tau)$ is uniform, the expected regret is just the expectation of a geometric random variable with probability $\frac{1}{n}$, which gives an expected regret of $n$. Meanwhile, the non-Markovian policy gets expected regret of $\frac{n}{2}$ (modulo off-by-one errors), so they differ by a factor of 2.

---

> > > > > ### Author Response · Authors · 2019-11-14
> > > > > **Why the settings we consider are reasonable**
> > > > >
> > > > > Thanks for the detailed response; it was helpful for clarifying our own thinking as well. Below, we aim to convince you that (1) Markovian policies _are_ optimal for a carefully defined meta-POMDP, and (2) the notion of adversarially robustness discussed in Section 5 does "generalize," in the sense that it affords robustness against other sorts of adversaries.
> > > > >
> > > > > *Meta-POMDP*
> > > > > There's a version of the meta-POMDP where memoryless policies are optimal. When defining the meta-POMDP, it is important to consider whether the agent observes when the meta-episode terminates. If the agent observes that the meta-episode terminates, then an optimal agent will iterate over possible trajectories $\tau$ in order of their density under $p(\tau)$ and restart the processes whenever the target trajectory $\tau^*$ is found. If, however, the agent does not observe when the meta-episode terminates, then this is not the optimal policy; the agent never knows when to restart at $\arg\max_\tau p(\tau)$. The optimal for this meta-POMDP is given in Section 4.2. The reason a memoryless policy is optimal for this carefully defined meta-POMDP is because no feedback is ever given, so the agent has nothing to remember.* We opted to exclude this discussion in the paper because we feared it would be overly confusing, but could include it if you think it would strengthen the paper.
> > > > >
> > > > > We argue that this type of meta-POMDP is a reasonable model of many real-world settings. We give three examples:
> > > > > 1. A patient tells his physician that he is ill. The physician prescribes a drug. The next week, the patient returns to his physician and says that he is still ill. The physician reasons that either (1) the first drug did not cure the first illness, or (2) the first drug did cure the first illness but the patient came down with a second illness. The physician must then decide which drug to prescribe for the next week. This process continues for many weeks. In this setting, regret corresponds to the number of weeks that the patient is sick. This type of scenario may occur in cancer treatment, where one drug may initially treat the cancer but suppress a patient's immune system and thus make him susceptible to other diseases.
> > > > > 2. An ecologist wants to increase the salmon population in the Klamath River. Each year, the ecologist can decide to perform one of many treatments, such as restricting fishing, increasing the flow from upstream dams, or seeding the river with salmon larvae. The next year the ecologist returns and finds that the fish population remains the same. The ecologist reasons that either (1) the treatment was effective in increasing the salmon population but some other factor (e.g., increased sewage runoff) caused a subsequent decline, or (2) the treatment was not effective. The ecologist must then decide whether to repeat the treatment from the first year or to try a new treatment. In this setting, regret corresponds to the number of years where the salmon population is below some sustainable level.
> > > > > 3. A school teacher hears a rumor that some students are shoplifting. The teacher can choose to talk with students to persuade them that shoplifting is bad. Obviously, none of the students will admit to the teacher that they shoplift. If the teacher talks with a student who shoplifted, the student will (unbeknownst to the teacher) return the stolen items. The student may still shoplift in the future. The teacher never learns which students shoplift, and must use his prior belief about who may shoplift to choose which students to talk with every day. In this setting, regret is to the average time before a stolen item is returned.
> > > > >
> > > > > *Adversarial Robustness*
> > > > > In Section 5, we showed that MaxEnt RL was optimal for adversarial robustness against an adversary who chooses a reward function from a carefully defined set of reward functions (Eq 2). Does robustness against this adversary provide robustness against other adversaries? Yes. In Appendix B.6, Eq 17 converts an arbitrary set of reward functions into a set of reward functions of the type that MaxEnt RL is robust against. We then use MaxEnt RL to find the optimal robust policy for this new set of reward functions. Finally, we evaluate the policy found by MaxEnt RL for adversarial robustness against the _original_ set of reward functions. While, in theory (Corollary B.4.1) the policy obtained by MaxEnt RL is only a lower bound on the optimal minimax policy for the original set of rewards, empirically we find that it is nearly optimal: as shown in Fig 7, the "generalization gap" between the MaxEnt RL policy and the optimal minimax policy is quite small (9%).
> > > > >
> > > > > Footnote:
> > > > > * One potential criticism of this setting is that, since no feedback is ever given, it is impossible to learn the optimal policy by interacting with the environment. However, the reward function in Lemma 4.1 does provide feedback to the agent, and does so in such a way that the optimal MaxEnt RL policy is optimal for the meta-POMDP.

---

> > > > > > ### Comment · AnonReviewer3 · 2019-11-14
> > > > > > **The stochastic policy is not optimal for the meta-POMDP**
> > > > > >
> > > > > > In general, if you allow the full set of policies, and there are no adversaries in the environment (i.e. the environment “response” doesn’t depend on your _policy_, and only on your actual action), then there is always an optimal deterministic policy. It can also be the case that there are optimal stochastic policies, but they are never required.
> > > > > >
> > > > > > I’m not totally sure what the new meta-POMDP is -- given my understanding, the stochastic policy is not optimal for the meta-POMDP.
> > > > > >
> > > > > > If the goal trajectory $\tau^*$ is resampled from $p(\tau^*)$ at every meta-timestep (or equivalently the meta-episode is always of length 1), then the optimal policy is to always deterministically play the most likely trajectory $\arg \max_{\tau^*} p(\tau^*)$.
> > > > > >
> > > > > > My best guess is that what you actually mean is that the meta-episodes remain the same -- they continue as long as the agent plays the wrong trajectory, and as soon as the agent plays the right trajectory, the meta-episode ends, and a new goal trajectory $\tau^*$ is sampled, and a new meta-episode begins -- but the agent doesn’t know this.
> > > > > >
> > > > > > In this case, from the agent’s perspective, it’s simply one very long episode where every time it plays the right goal trajectory, the environment “transitions” to a new state with a new goal trajectory. This is a (non-meta) POMDP, and we can solve it by maintaining a belief over the hidden state. Specifically, if our current belief over the goal trajectory is $b(\tau^*)$, and we play $\tau$, then our new belief is:
> > > > > > $$b’(\tau^* \mid \tau) = b(\tau) p(\tau^*) + 1[\tau^* \neq \tau] b(\tau^*)$$
> > > > > > (The first term comes from the chance that $\tau$ was the goal trajectory, and the second term is if $\tau$ is not the goal trajectory.)
> > > > > > The optimal policy is then:
> > > > > > $$\pi(\tau \mid b(\tau^*)) = \arg \max_{\tau^*} b(\tau^*)$$
> > > > > >
> > > > > > This policy can do much better than the policy of Section 4.2. For example, consider the case where we are uncertain between two trajectories: $p(\tau_1) = p(\tau_2) = 0.5$. Then the Section 4.2 policy is to choose randomly between $\tau_1$ and $\tau_2$, which incurs an expected $\frac{1}{2}$ regret per meta-timestep.
> > > > > >
> > > > > > In contrast, the policy above recommends always switching between $\tau_1$ and $\tau_2$, that is, it plays $\tau_1, \tau_2, \tau_1, \tau_2, \dots$. In this case, you can verify that in the limit of many timesteps, $b(\tau^*)$ switches between ${\tau_1 : \frac{2}{3}, \tau_2 : \frac{1}{3}}$ and ${\tau_1: \frac{1}{3}, \tau_2: \frac{2}{3}}$. Thus, it gets an expected $\frac{1}{3}$ regret per meta-timestep.
> > > > > >
> > > > > > Just to confirm, here’s a Python simulation:
> > > > > > ```
> > > > > > from random import seed, randint
> > > > > >
> > > > > > def simulate(n):
> > > > > >     regret = 0
> > > > > >     goal = randint(0, 1)
> > > > > >     for action in [0, 1] * (n // 2):
> > > > > >         if goal == action:
> > > > > >             # Correct trajectory, no regret, resample goal
> > > > > >             goal = randint(0, 1)
> > > > > >         else:
> > > > > >             # Wrong trajectory, add regret, don't resample goal
> > > > > >             regret += 1
> > > > > >     return regret
> > > > > >
> > > > > > seed(0)
> > > > > > regrets = [simulate(1000) for _ in range(10)]
> > > > > > average_regret = float(sum(regrets)) / len(regrets)
> > > > > > print("Regrets for 1000 timesteps: {}\nAverage regret: {}".format(regrets, average_regret))
> > > > > > ```
> > > > > > This gives me the results:
> > > > > > ```
> > > > > > Regrets for 1000 timesteps: [343, 321, 329, 336, 345, 329, 339, 335, 351, 331]
> > > > > > Average regret: 335.9
> > > > > > ```
> > > > > >
> > > > > > Overall, the broader point is that in the absence of adversaries and limiting assumptions like a Markovian policy, there is always an optimal deterministic policy. Intuitively, for MDPs, a stochastic policy is a mixture of deterministic policies, and in the absence of adversaries, the expected reward of a mixture is a convex combination of the expected rewards of the individual components, and so if the stochastic policy is optimal, all of the component deterministic policies must also be optimal. For POMDPs, the same applies of the resulting MDP over belief states. This critique didn’t initially apply because by forcing the policy to be Markovian over regular states, you excluded deterministic policies over belief states. Unfortunately I don’t have a reference for all of this, but such a reference probably exists.
> > > > > >
> > > > > > Since there is always an optimal deterministic policy, I don’t expect that an analysis of the form “MaxEnt RL optimally solves this POMDP” would be able to justify MaxEnt RL over regular RL.

---

> > > > > > > ### Author Response · Authors · 2019-11-15
> > > > > > > **Thanks for the correction.**
> > > > > > >
> > > > > > > Thanks for the detailed correction -- we hadn't noticed that policies with memory achieved lower regret in this setting. We stand by our previous statement that MaxEnt RL is optimal among memoryless policies.
> > > > > > >
> > > > > > > Broadly, I think we agree that MaxEnt RL is _not_ optimal for maximizing reward in an MDP. As you noted above (and we noted in Section 2, paragraph 2), MDPs always have deterministic solutions. Nonetheless, MaxEnt RL algorithms seem adept at solving standard MDPs. This should strike you as quite odd -- why should optimizing objective A yield good performance on objective B? We do not have a full answer to this question, but believe that this question will be of interest to the ICLR community. We believe that the analysis in this paper provides a set of useful tools and mental models for studying this question, and will catalyze more work in understanding why MaxEnt RL works.

---

> > > > > > ### Comment · AnonReviewer3 · 2019-11-14
> > > > > > **Robust control problems seem interesting to analyze**
> > > > > >
> > > > > > I like Appendix B, and I’d recommend turning that into a paper, and getting rid of the parts about meta-POMDPs (see other comments). Given that it’s nearly a new paper, I’d want it to go through a new peer review process, but here are some quick comments:
> > > > > >
> > > > > > 1a. Most of the theorems ultimately come down to the fact that the reward $\log \pi(a \mid s)$ perfectly encodes the policy in a manner that can be reconstructed by MaxEnt RL. This is something that regular RL cannot do, because with regular RL there is always an optimal deterministic policy, and so no reward can enforce that regular RL produce a unique optimal stochastic policy. (Though as Appendix B.9 notes, regular RL can have _an_ optimal stochastic policy.) However, I wouldn’t be surprised if there are many other procedures that take as input a function $S \times A \rightarrow \mathbb{R}$ and produce as output a function $S \times A \rightarrow \mathcal{D}(A)$ that also have similar theorems, where you can say that there exists some input that produces any particular output (i.e. the procedure is surjective). What distinguishes MaxEnt RL from these other procedures?
> > > > > >
> > > > > > 1b. “Does robustness against this adversary provide robustness against other adversaries? Yes.” The way I would phrase the argument is “for any adversary, there is some best policy for that adversary, and that policy can be encoded into a reward function that MaxEnt RL can turn back into the same policy”. This argument applies to any surjective function from rewards to policies, not just MaxEnt RL. The real question is whether, _for the actual rewards we use MaxEnt RL with_, the resulting policies are robust to a broad range of adversaries.
> > > > > >
> > > > > > 2. To what extent can these robust control results be extended to regular RL? They won’t all generalize, because regular RL doesn’t lead to unique stochastic optimal policies, which are necessary for some robust control settings, but some will -- for example, there’s an equivalent of Lemma B.1-B.3 for regular RL.
> > > > > >
> > > > > > 3. The results with the temperatures seem to cut against the claim that MaxEnt RL is leading to robustness. While I agree that too much robustness is a bad thing, one would expect that there is some “sweet spot” of the right amount of robustness. With temperatures, as the temperature is decreased to zero, the robust reward set grows larger and larger, with the largest set being with temperature zero, i.e. regular RL. (It’s worth noting that this set is just the boring set $\{ r \}^{+}$, using the notation of Lemma B.1.) While this would naively suggest that regular RL is more robust than MaxEnt RL, I think it really suggests that analyzing the “size” of the robust reward set is not meaningful, which makes sense since the invariances in reward space (affine transformations) mean that regular R^n reasoning about it tends to give misleading conclusions.
> > > > > >
> > > > > > 4. For Section B.6, it is probably worth noting that by making r* arbitrarily negative, the constraints can be satisfied, and so the set $R^*(R)$ is non-empty. I didn’t immediately realize that.
> > > > > >
> > > > > > 5. I don’t buy the proof of Theorem B.5. The problem is that it is not possible to find a policy with a given occupancy measure $\rho(s, a)$ if the adversary can control the _transition dynamics_. The theorem in Ziebart (2010) doesn’t deal with this case. I suspect the theorem is true regardless -- since the adversary can only control things that are Markovian, you should only need mixtures over actions given states to be robust, which means you only need a single stochastic policy.

---

> > > > > > > ### Author Response · Authors · 2019-11-15
> > > > > > > **Responses to questions**
> > > > > > >
> > > > > > > Thanks for all the additional feedback on Appendix B!
> > > > > > >
> > > > > > > 1a Perhaps the most important distinguishing factor is that we have efficient algorithms for MaxEnt RL. For example, current MaxEnt RL algorithms don't involve belief state estimation (as is necessary for solving many POMDPs) or solving a two-player game (as is necessary for many adversarial settings).
> > > > > > >
> > > > > > > 1b. To clarify, the procedure in Appendix B.6 and experiment in Appendix B.7 are in a slightly different setting. While it is true that MaxEnt RL allows any policy to be converted into a reward function and vice-versa, the proofs of this fact that we give are not constructive, in the sense that constructing the corresponding reward function requires knowing the optimal policy apriori. The procedure in Appendix B.6 aims to lift this limitation, providing a reward function to which we can apply MaxEnt RL, _without_ requiring knowledge of the optimal policy. The experiment in Appendix B.7 shows that this (approximate) procedure works well.
> > > > > > >
> > > > > > > 2. While the groundwork we lay for discussing robust-reward control (e.g., Lemmas B.1 - B.3) is not specific to MaxEnt RL, to the best of our knowledge the "useful" results (Thm 5.2 and Corollary B.4.1) apply uniquely to MaxEnt RL.
> > > > > > >
> > > > > > > 3. We agree that the "size" of the set is somewhat confusing, which motivated us to visualize temperatures in two ways in Figure 6, depending on how you frame the question (what is the size of the robust set for a given MaxEnt RL reward function, or what is the size of the robust set that _includes_ a given MaxEnt RL reward function).  One way to automatically tune the temperature parameter might be to introduce the temperature as an additional parameter into the optimization problem in Eq 17.
> > > > > > >
> > > > > > > 4. We have updated the paper to incorporate this suggestion.
> > > > > > >
> > > > > > > 5. Thanks for catching this! We will clarify this proof for the final version of the paper.

---

### Decision · Program_Chairs · 2019-12-19

**Decision:**

Reject

**Comment:**

This paper studies maximum entropy reinforcement learning in more detail. Maximum entropy is a popular strategy in modern RL methods and also seems used in human and animal decision making. However, it does not lead to optimize expected utility. The authors propose a setting in which maximum entropy RL is an optimal solution.

The authors were quite split on the paper, and there has been an animated discussion between the reviewers among each other and with the authors.

The technical quality is good, although one reviewer commented on the restricted setting of the experiments (bandit problems). The authors have addressed this by adding an additional experiment. Futhermore, two reviewers commented that the clarity of the paper could be improved.

A larger part of the discussion (also the private discussion) revolved around relevance and significance, especially of the meta-pomdp setting that takes up a large part of the manuscript.
- A reviewer mentioned that after reading the paper, it does not become more clear why maximum entropy RL works well in practice. The discussion even turned to why MaxEntropyRL might be *unreasonable* from the point of view of needing a meta-POMDP with Markov assumptions, which doesn't help shed light on its empirical success. The meta-POMDP setting does not seem to reflect the use cases where maximum entropy RL has done well in emperical studies.
- Another reviewer mentioned that earlier papers have investigated maximum entropy RL, and that the paper tries to offer a new perspective with the Meta-POMDP setting. The discussion of this discussion was not deemed complete in current state and needs more attention (splitting the paper into two along these lines is a possibility mooted by two of the reviewers). A particular example was the doctor-patient example, where in the meta-POMDP setting the doctor would repeatedly attempt to cure a fixed sampled illness, rather than e.g. solving for a new illness each time.

Based on the discussion, I would conclude that the topic broached by the paper is very relevant and timely, however, that the paper would benefit from a round of major revision and resubmission rather than being accepted to ICLR in current form.